# Motion Control of High-Dimensional Musculoskeletal Systems with Hierarchical Model-Based Planning

**Yunyue Wei[1], Shanning Zhuang[1], Vincent Zhuang[2], Yanan Sui[1]**
[1] Tsinghua University [2] Google DeepMind
`{weiyy20, zhuangsn24}@mails.tsinghua.edu.cn`
`vincentzhuang@google.com`
`ysui@tsinghua.edu.cn`

## Abstract

Controlling high-dimensional nonlinear systems, such as those found in biological and robotic applications, is challenging due to large state and action spaces. While deep reinforcement learning has achieved a number of successes in these domains, it is computationally intensive and time consuming, and therefore not suitable for solving large collections of tasks that require significant manual tuning. In this work, we introduce Model Predictive Control with Morphology-aware Proportional Control ($\text{MPC}^2$), a hierarchical model-based learning algorithm for zero-shot and near-real-time control of high-dimensional complex dynamical systems. $\text{MPC}^2$ uses a sampling-based model predictive controller for target posture planning, and enables robust control for high-dimensional tasks by incorporating a morphology-aware proportional controller for actuator coordination. The algorithm enables motion control of a high-dimensional human musculoskeletal model in a variety of motion tasks, such as standing, walking on different terrains, and imitating sports activities. The reward function of $\text{MPC}^2$ can be tuned via black-box optimization, drastically reducing the need for human-intensive reward engineering.

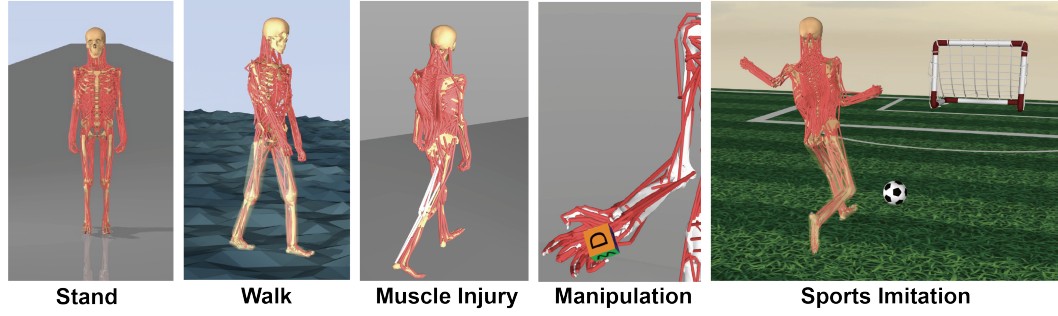

| Stand | Walk | Muscle Injury | Manipulation | Sports Imitation |

Figure 1: **Movement control of whole-body human musculoskeletal system over a diverse set of motion control tasks.** The videos of the control performances are on the **project page**.

## 1 Introduction

High-dimensional nonlinear dynamical systems are prevalent in the real world, with important examples including biological musculoskeletal systems. The system complexity laid the foundation of flexible motion due to their over-actuated nature. The presence of redundant actuation enhances the safety and robustness of the system, reducing the risk of performance degradation from actuator faults (Hsu et al., 1989). However, it also leads to large state and action spaces, posing significant challenges to achieving stable control performance. We take the human musculoskeletal system as

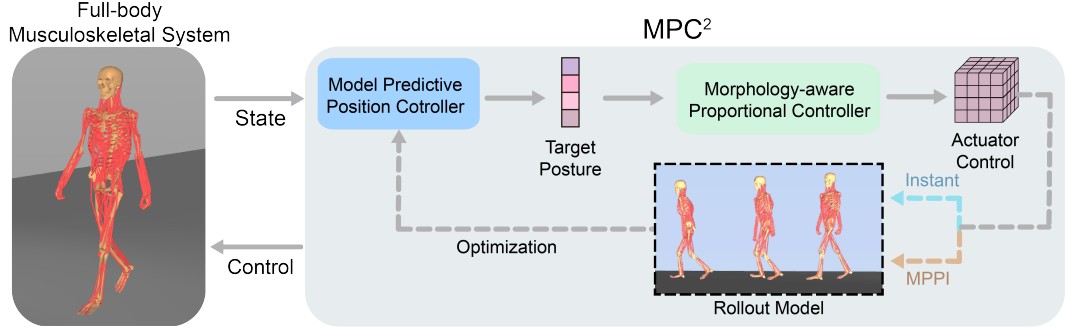

Figure 2: Workflow of Model Predictive Control with Morphology-aware Proportional Control (MPC$^2$). Solid arrows indicate control pipeline, and dashed arrows indicate planning procedure.

a key example, where hundreds of muscles coordinate to facilitate various movements. Understanding and optimizing control in such systems is crucial for applications in healthcare and human robot interaction (Kidziński et al., 2018; Vittorio et al., 2022).

Deep reinforcement learning (DRL) is a promising approach for controlling high-dimensional systems. However, RL approaches often struggle in high-dimensional state and action spaces, and typically require the use of lower-dimensional representations. More importantly, the immense computational requirements of DRL imposes a severe bottleneck on the iteration speed of reward engineering, meaning that researchers often need days (or longer) to discover effective control policies. Furthermore, the high nonlinearity of the musculoskeletal system necessitates considerable sequential computation, complicating its parallel deployment on GPUs and impeding the training speed of DRL algorithms. Being able to generate effective control policies for high-dimensional nonlinear dynamical systems in *near real-time* is an open challenge.

Clinical studies on motor control of human movement revealed that predictive sampling is a crucial strategy in human movement control, such as maintaining balance during walking (Winter, 1991; Patla, 2003), where planning over a finite horizon determines the controls to be executed. Recent works have started to incorporate model predictive control (MPC) as the control backbone, offering faster behavior synthesis and more efficient reward design compared to DRL (Howell et al., 2022; Yu et al., 2023). However, effective planning in high-dimensional control spaces remains challenging, limiting the success of MPC primarily to low-dimensional systems. To the best of our knowledge, no training-free methods have achieved stable movement control of a whole-body musculoskeletal model across varying task conditions.

In this paper, we propose Model Predictive Control with Morphology-aware Proportional Control (MPC$^2$), a hierarchical model-based planning algorithm designed to address the challenges of high-dimensional musculoskeletal control. We introduce a sampling-based model predictive controller to plan the target posture of the agent, while a morphology-aware proportional controller serves as the low-level policy, adaptively coordinating the actuators to achieve the target joint positions. We demonstrate that our method can achieve stable control of a 700-actuator whole-body musculoskeletal model *without training*, enabling tasks such as standing, walking over varying terrain conditions, and sports motion imitation (Figure 1). Furthermore, we show that MPC$^2$'s fast control generation facilitates efficient cost function optimization, improving task performance, especially for performing complex sequences of movement. The bottleneck in achieving *real-time* control with MPC$^2$ is the speed of the additional model forward dynamics computation, which can be solved by using more powerful computing devices or by controlling systems with reduced complexity.

**Our contributions.** (1) We propose MPC$^2$, the first MPC-based method capable of achieving near real-time stable control of high-dimensional musculoskeletal systems. (2) We demonstrate that our hierarchical model predictive control algorithm enables zero-shot high-dimensional full-body motion control across a wide range of motion tasks, many of which have not been achieved by state-of-the-art DRL-based methods. (3) We show that the much faster control generation latency of MPC$^2$ facilitates automated cost function optimization via Bayesian optimization, demonstrating a pathway for reducing the human burden of reward engineering to near zero.

## 2 RELATED WORK

**High-dimensional musculoskeletal control.** The control of musculoskeletal systems is challenging due to both high dimensionality and non-linearity, with deep reinforcement learning (DRL) being the predominant choice in existing solutions (Kidziński et al., 2018; Geiß et al., 2024). Hierarchical architectures are often employed to decompose control across different modules, where DRL provides high-level actions and a low-level policy generates muscle controls (Lee et al., 2019; Park et al., 2022; Feng et al., 2023). These approaches typically require large collections of motion data for imitation learning. Several works have also explored strategies to improve sample efficiency in over-actuated regime, including bio-inspired exploration (Schumacher et al., 2022), latent space exploration (Chiappa et al., 2023), model-based planning (Hansen et al., 2023), and multi-task learning (Caggiano et al., 2023). Recent studies have leveraged muscle synergies to reduce control dimensionality, enabling stable control across various musculoskeletal models (Berg et al., 2023; He et al., 2024). These methods typically require many hours or days of training to achieve effective control, posing a significant bottleneck on the iteration speed of reward engineering.

**Model-predictive control.** Compared to DRL, model predictive control allows for real-time control (Tassa et al., 2012), and thus has seen an increasing application of MPC in robotics, including tasks such as quadruped locomotion and dexterous manipulation (Kim et al., 2023). Recent works have also integrated MPC into the reward design process due to its training-free nature (Jain et al., 2021; Yu et al., 2023; Liang et al., 2024). However, MPC typically succeeds only in low-dimensional settings and often struggles when applied to high-dimensional problems. The most complex systems handled by existing MPC-based methods are typically torque-driven humanoids (Meser et al., 2024).

## 3 PRELIMINARIES

### 3.1 MUSCULOSKELETAL SYSTEM CONTROL

**High-dimensional over-actuated system.** In this paper, we used musculoskeletal models as the target high-dimensional over-actuated system, where the model dynamics can be formulated as follows:

$$M(q)\ddot{q} + c(q, \dot{q}) = J_m^T f_m + J_c^T f_c + \tau_{ext}, \tag{1}$$

where $q$ denotes generalized coordinates of joints, $M(q)$ denotes the mass distribution matrix, and $c(q, \dot{q})$ denotes Coriolis and the gravitational force applied over the generalized coordinates, $J_m$ and $J_c$ denote Jacobian matrices that map forces to the generalized coordinates, $f_c$ is the constraint force, $f_m$ denotes actuator forces, and $\tau_{ext}$ denotes all external torque when interacting with environments.

Our used musculoskeletal models are implemented in the MuJoCo physics simulator (Todorov et al., 2012), where actuators are modeled as first-order systems. The force generated by one actuator can be formulated as follows:

$$f_m = F_k \cdot a + F_p, \quad \frac{\partial a}{\partial t} = \frac{u - a}{(u - a)\tau_1 + \tau_2}, \tag{2}$$

where $a$ is the actuator activation, $F_k, F_p$ represents the gain and bias of the actuator force dynamics, $u$ denotes the actuator control, $\tau_1$ and $\tau_2$ denote the time coefficients of the first-order actuator system. Our primary experiments are conducted on the MS-Human-700, a comprehensive whole-body musculoskeletal model comprising 90 rigid body segments, 206 joints, and 700 muscle-tendon units (Zuo et al., 2024). Additionally, we employ an upper limb model of the human body and an ostrich model (La Barbera et al., 2021) to showcase the generalization of $\mathsf{MPC}^2$ across different models and tasks.

**Problem formulation.** We treat the high-dimensional over-actuated control problem as a finite horizon Markov decision process with state $s \in \mathcal{S}$, control $u \in \mathcal{U}$, and dynamics $f$. For a given initial state of the model $s_0$ and a desired horizon $T$, we aim to find a control sequence $\boldsymbol{u}_{0:T}^{\star} = (u_0, ..., u_{T-1})$ that enable stable control, which can be achieved by minimizing the cumulative value of a task-specific cost function $C_\theta$ parameterized by $\theta$:

$$\boldsymbol{u}_{0:T}^{\star} = \operatorname{argmin}_{\boldsymbol{u}_{0:T}} \sum_{t=0}^{T-1} C_\theta(s_t, u_t), s_{t+1} = f(s_t, u_t) \tag{3}$$

In this paper, the definition of cost function $C_\theta$ is equivalent to the reward functions used in reinforcement learning (with negative value for maximization). For MS-Human-700, we consider the action space is $d_u = 700$-dimensional control of actuators (muscle-tendon units). The state space of the full-body model consist of joint positions and velocities, actuator activations and lengths, and task-related observations, leading to space dimensionality $d_s$ over 1500.

## 3.2 SAMPLING-BASED MODEL PREDICTIVE CONTROL

Model predictive control is a general framework for model-based control, which optimizes a local control sequence using an approximated dynamics $\hat{f}$ within a short horizon $H \ll T$:

$$\hat{\boldsymbol{u}}_{t:t+H}^\theta = \text{argmin}_{\hat{\boldsymbol{u}}_{t:t+H}} \sum_{h=0}^{H-1} C_\theta(s_{t+h}, \hat{u}_{t+h}), s_{t+h+1} = \hat{f}(s_{t+h}, \hat{u}_{t+h}). \quad (4)$$

The optimized action sequence $\hat{\boldsymbol{u}}_{t:t+H}^\theta = (\hat{u}_t^\theta, \cdots, \hat{u}_{t+H-1}^\theta)$ is a local approximation of optimal controls $\boldsymbol{u}_{t:t+H}^\star$. In real-world deployment where the action execution and planning are asynchronous, the planning horizon $H$ should be chosen to balance accuracy and instantaneity.

Among various implementations of MPC frameworks, sampling-based MPC is a popular choice which samples local control sequences from a distribution of open-loop control sequences, $\hat{\boldsymbol{u}}_{t:t+H} \sim \boldsymbol{p}_\phi(\cdot)$, and update the sample distribution via parallel rollouts of the sampled action sequences. The objective of sampling-based MPC is to find a distribution parameter $\phi$ that minimize the cumulative cost function value of sampled action sequences. The distribution update process usually only depends on the rollout performance without direct operation on the states, which has been demonstrated success in the control of high degree-of-freedom systems, such as torque-driven humanoid models (Meser et al., 2024).

Model Predictive Path Integral (MPPI) control (Williams et al., 2016) is a commonly used sampling-based MPC method, which assumes the sampling distribution is a factorized Gaussian with $\phi = (\mu_t, \cdots, \mu_{t+H-1}, \sigma_t, \cdots, \sigma_{t+H-1})$:

$$\boldsymbol{p}_\phi(\hat{\boldsymbol{u}}_{t:t+H}) = \prod_{h=0}^{H-1} \mathcal{N}(\hat{u}_{t+h}; \mu_{t+h}, \sigma_{t+h}). \quad (5)$$

During the rollout process, $N$ action sequences $\{\hat{\boldsymbol{u}}_{t:t+H}\}_{n=1}^N$ are sampled and executed via approximated transition $\hat{f}$. For each sampled sequence $\hat{\boldsymbol{u}}_{t:t+H}^n$, the cumulative cost function $\mathcal{C}_\theta^n = \sum_{h=0}^{H-1} C_\theta(s_{t+h}, \hat{u}_{t+h}^n)$ is collected and used for distribution update:

$$\mu_{t+h} = \frac{\sum_{n=1}^N w_n \cdot \hat{u}_{t+h}^n}{\sum_{n=1}^N w_n}, \sigma_{t+h} = \sqrt{\frac{\sum_{n=1}^N w_n \cdot (\hat{u}_{t+h}^n - \mu_{t+h})^2}{\sum_{n=1}^N w_n}}, \quad 0 \le h \le H-1, \quad (6)$$

where $w_n = \mathbb{1}_{r(n) \le m} e^{-\frac{1}{\lambda} \mathcal{C}_\theta^n}$, $r(n)$ is the increasing-order rank of cumulative cost function value of rollout $n$, $m$ is the number of elite rollouts, and $\lambda$ is the temperature parameter.

## 3.3 OPTIMAL COST FUNCTION DESIGN

The finite-horizon optimization of MPC can result in a myopic policy, which may be suboptimal when evaluated in the long term. Recent studies demonstrate that the parameters of the cost function can be optimized to compensate for the issues induced by local optimization, which can be different from the true cost function measured in the full horizon $T$ (Jain et al., 2021; Le & Malikopoulos, 2023). The objective of optimal cost function design is to find parameters $\theta^*$ that minimizes the cumulative value of true cost function $C_\theta$ over the horizon $T$:

$$\theta^* = \text{argmin}_{\theta'} \sum_{t=0}^{T-1} C_\theta(s_t, \hat{u}_t^{\theta'}), s_{t+1} = f(s_t, \hat{u}_t^{\theta'}), \quad (7)$$

where $(\hat{u}_0^{\theta'}, \cdots, \hat{u}_H^{\theta'} - 1)$ is the control sequence of MPC optimized under cost function parameterized by $\theta'$. As only zero-order cost function value can be accessed, e.q. 7 can be considered as a black-box optimization problem, which can be addressed by Bayesian optimization or evolutionary algorithms.

---

**Algorithm 1:** Model Predictive Control with Morphology-aware Proportional Control ($\mathsf{MPC}^2$)

---

**Input:** Model dynamics $f$, rollout horizon $H$, total rollout number $N$, instant rollout number $\bar{N}$, iteration number $r$, distribution parameter $\mu, \sigma$, current state $s_t$

1 **for** $i = 1, \cdots, r$ **do**
2     $z^1, ..., z^{\bar{N}} \sim \mathcal{N}(M_{\text{pos}}(s_t), \sigma)$                         // Instant rollout
3     $z^{\bar{N}+1}, ..., z^N \sim \mathcal{N}(\mu, \sigma)$                            // MPPI rollout
4     $\mathcal{C}_\theta^1, \cdots, \mathcal{C}_\theta^N \leftarrow \mathcal{R}_{\text{MP}}(z^1, H), \cdots, \mathcal{R}_{\text{MP}}(z^N, H)$
5     Update $\mu, \sigma$ using e.q. (6)
6 **end**
7 $z^* \leftarrow \mu, \quad \hat{u}_t^\theta \leftarrow \pi_{\text{MP}}(s_t, z^*)$
8 **return** $\hat{u}_t^\theta, z^*, \mu, \sigma$

---

# 4    MODEL PREDICTIVE CONTROL WITH MORPHOLOGY-AWARE PROPORTIONAL CONTROL ($\mathsf{MPC}^2$)

Existing approaches for controlling high-dimensional musculoskeletal systems often incorporate deep reinforcement learning as a central component, where a state-feedback policy, $\pi(u|s)$, is learned from interactions with the model dynamics. While substantial efforts have been made to reduce the dimensionality of the action space, the large state space continues to present significant challenges for policy training. In this paper, we opt to use model predictive control instead of deep reinforcement learning for the following reasons: 1) the overall control is conducted in simulation, where the exact dynamics is accessible, that is $\hat{f} = f$; 2) the use of sampling-based MPC circumvents the challenge of decision-making in high-dimensional state spaces; 3) MPC offers much faster control generation, enabling more reward design optimization iterations than DRL.

However, directly deploying MPC on musculoskeletal systems is challenging because of the high dimensionality. In this section, we demonstrate that applying MPC to such problems is indeed possible. Our approach is motivated by the observation that many biological systems such as vertebrates utilize hierarchical control strategies, in which sensory information is processed by a high-level controller for planning, while motor commands are generated by a low-level controller based on proprioception (Merel et al., 2019). To this end, we introduce $\mathsf{MPC}^2$, a hierarchical MPC method that facilitates stable control of high-dimensional musculoskeletal systems, as shown in Figure 2 and Algorithm 1. $\mathsf{MPC}^2$ has two major components: (1) a model predictive position controller as the high-level planner which optimize for the target posture $z^*$ given current state $s_t$; and (2) a morphology-aware proportional controller $\pi_{\text{MP}}(u|s, z)$ as the low-level policy which computes actuator controls to achieve the target posture from given state.

## 4.1   MODEL PREDICTIVE POSITION CONTROL

We employ MPC over the planning of major joint coordinates $z$ that determine the system posture. For MS-Human-700, the dimension of $z$ is $d_z = 37$. Compared to torque, we choose lower-order joint position as the MPC objective to reduce the control frequency. Therefore, only one target posture $z$ is required to optimize during one rollout, where our morphology-aware proportional controller adapts control signals based on the instant states:

$$\mathcal{C}_\theta = \mathcal{R}_{\text{MP}}(z, H) = \sum_{h=0}^{H-1} C(s_{t+h}, u_{t+h}), u_{t+h} = \pi_{\text{MP}}(s_{t+h}, z). \tag{8}$$

Compared to planning over original action space, $\mathsf{MPC}^2$ significantly reduce the number planning parameters from $H \cdot d_u$ to $d_z$, enabling optimizing controls via sampling. While the original Model Predictive Path Integral (MPPI) can be directly employed as a high-level planner for target positions, we find that it lacks the ability to respond quickly to rapidly changing states, such as when the agent is falling. This issue cannot be easily mitigated by simply increasing the number of rollouts, as only a limited number (at most a few dozen) can be executed in parallel when controlling high-dimensional systems, due to both computational budget constraints and the need for real-time responsiveness.

To equip MPC with rapid response capabilities for changing states, we leverage the feature of position control and propose the use of *instant rollouts* during planning (line 2 in Algorithm 1). Rather than sampling based on the policy from the previous planning iteration, instant rollouts sample target postures based on the model's current posture, which can be extracted from the current state using a posture mask: $z_t = M_{\text{pos}}(s_t)$. When the current state significantly deviates from the previous planning state, this approach provides a better initial point compared to the original MPPI samples, increasing the likelihood of sampling more effective controls to re-stabilize the agent. We will demonstrate the necessity of instant rollouts for position control in the experimental section.

## 4.2 MORPHOLOGY-AWARE PROPORTIONAL CONTROL

Here we introduce the morphology-aware proportional controller $\pi_{\text{MP}}(u|s, z)$, a key component for reducing the control dimensionality, which coordinates actuator controls to adapt to the target posture. Given the target joint coordinate $z^*$, the target actuator length $l^*$ can be computed with model forward dynamics. We define proportional controllers for each actuator, which determine the actuator force required to achieve the target actuator length given current actuator length $l$:

$$f_m^* = \min(0, k \cdot (l^* - l)), \tag{9}$$

where $k$ is the proportional gain parameter. Utilizing the first-order actuator dynamics in 2, we are able to derive the control signal $u^*$ to achieve target actuator force $f_m^*$ given current actuator activation $a$:

$$u^* = a + \frac{\tau_2(a^* - a)}{\Delta t - \tau_1(a^* - a)}, \tag{10}$$

where $a^* = (f_m^* - F_p)/F_k$ is the target actuator activation, and $\Delta t$ is the duration of each time step. The proportion gain vector $K = (k_1, \cdots, k_{d_u})$ controls the scaling of target forces, which is critical for the control performance. Improper gain settings can result in excessive collisions (if too large), insufficient force generation (if too small), which should be individually set for each of the 700 actuators.

From system dynamics in e.q. 1, the conversion from actuator forces to joint torque is computed using the Jacobian matrices of the model, $J_m$, which can represent the influence of actuators on joint movements. Based on this observation, we propose to set proportional gains according to the system morphology. Instead of manually setting these gains, we set them based on the Jacobian matrices of current state and the target posture:

$$K = \bar{k} \cdot \sum_{i \in \mathcal{I}_z} |\text{col}_i(J_m) \cdot [z_i^* - M_{\text{pos}}(s_t)_i]|, \tag{11}$$

where $\bar{k}$ is the only scaling parameter, $\mathcal{I}_z$ is the indices of major joints $z$ over all joints, $|\cdot|$ is the absolute value operator, and $\text{col}_{(\cdot)}(J_m)$ is the column operator of $J_m$. The Jacobian values vary according to different system posture, allowing for adaptive and efficient control of different motion.

Note that $\text{MPC}^2$ achieves high-dimensional musculoskeletal control through online planning using model dynamics, allowing for the control of complex behaviors without the need for a training procedure. This zero-shot motion control also enables rapid evaluation of cost function designs, facilitating efficient optimization of the cost function.

## 5 EXPERIMENTS

In this section, we aim to comprehensively evaluate $\text{MPC}^2$, and seek to answer the following questions: 1) Can $\text{MPC}^2$ achieve robust and performant control over a wide variety of motion tasks and models? 2) Can $\text{MPC}^2$ generalize across different model morphology? 3) Can we leverage the fast generation speed of $\text{MPC}^2$ to serve as the inner loop in a reward function optimization problem? 4) How do the individual components of $\text{MPC}^2$ contribute to its overall effectiveness?

**Implementation details.** We implement $\text{MPC}^2$ using the Mujoco MPC (MJPC) platform (Howell et al., 2022), a framework designed for real-time model predictive control. The MJPC platform supports asynchronous simulation between the main thread and planning, which we find to be more practical than freezing the main thread during planning. In all experiments, we set the iteration

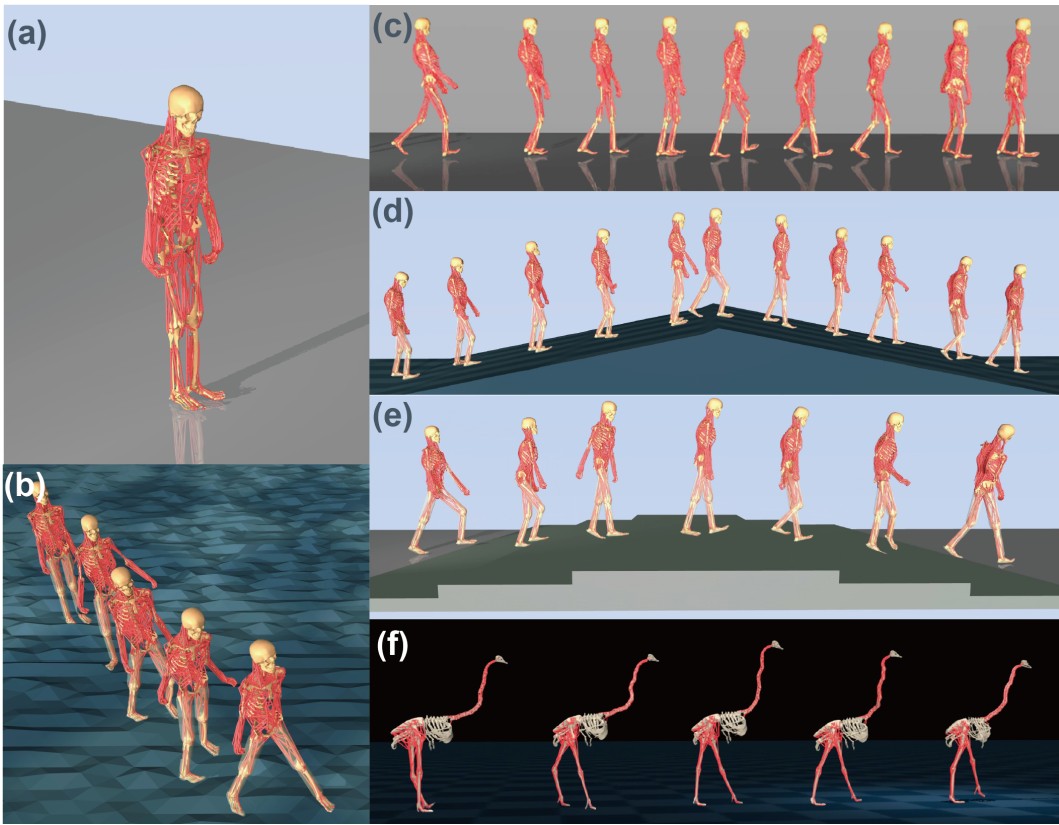

Figure 3: Control sequences of $\mathsf{MPC}^2$ in (a) Stand, (b) Rough, (c) Walk, (d) Slope and (e) Stair (f) Ostrich walk tasks. The simulation speed of Stair task is set to 10% due to slower contact computation.

number $r$ of $\mathsf{MPC}^2$ to 1 for rapid response to the changing states in the main thread, and sample 64 rollouts (containing $\bar{N} = 10$ instant rollouts) across a 0.3s horizon during each round of planning.

Unless otherwise noted, the simulation in main thread are run with 20% of the real-time speed (following Howell et al. (2022)), where control sequences to complete the task can be generated within 2 minutes. The experiments of $\mathsf{MPC}^2$ were conducted on a server equipped with an AMD EPYC 7773X processor, an NVIDIA GeForce RTX 4090 GPU, and 512 GB of memory. The cost function design of all movement tasks is detailed in Appendix B.

## 5.1 ROBUST MOVEMENT CONTROL

**Motion control over different terrain.** We design the following control tasks, which consists of a wide range of human full-body motion: (1) **Stand**. This task requires standing still and keep balance for 10 seconds. (2) **Walk**. This task requires walking forward over a flat floor for 10 meters. (3) **Rough**. This task requires walking forward over a rough terrain for 10 meters. (4) **Slope** This task requires walking up and down slopes. (5) **Stair**. This task requires walking up and down stairs.

We show control sequences of the Stand, Walk, Rough, Slope, and Stair tasks using $\mathsf{MPC}^2$ in Figure 3. While no previous control methods have demonstrated success in whole-body musculoskeletal systems for these tasks, $\mathsf{MPC}^2$ exhibits consistent and stable control performance across various tasks, enabling navigation over different terrain conditions.

**Adaption to model changes.** We demonstrate that $\mathsf{MPC}^2$ effectively leverages the over-actuated nature of musculoskeletal systems to achieve stable control even in the presence of actuator failures. As shown in the Figure 4(a), we disabled the biceps femoris, gastrocnemius, semitendinosus, and semimembranosus muscles on the back of the right leg at the 5th second of walking to test whether $\mathsf{MPC}^2$ can continue to walk. This requires the controller to use the overdrive of the system to adap-

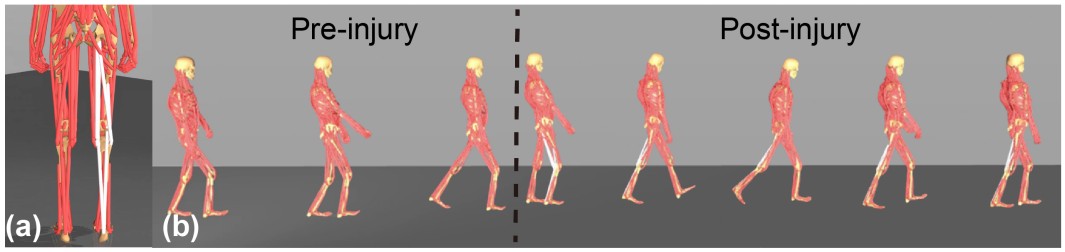

Figure 4: Walking control under sudden muscle failure. (a) Illustration of muscle injury. (b) Control sequences of $\mathsf{MPC}^2$. The dotted line shows the time when the muscles fail.

tively adjust the control strategy. As illustrated in Video W10 and Figure 4(b), $\mathsf{MPC}^2$ dynamically adapts its control strategy to maintain forward walking despite the sudden disablement of the posterior muscles in the right leg. We also find trained DRL agent fails to walk with actuator faults, as shown in the Video W3.

**Robustness with respect to perturbation forces.** The $\mathsf{MPC}^2$ algorithm demonstrates robustness to certain perturbations, as evidenced in two walking scenarios. As shown in Video W11 and W12, $\mathsf{MPC}^2$ successfully maintains forward walking despite the application of significant external forces, including large, random, short-term forces (500N applied for 0.2 seconds every 1 second) and consistent, random forces (100N applied continuously). These results highlight the system's ability to adapt and maintain stability under challenging conditions.

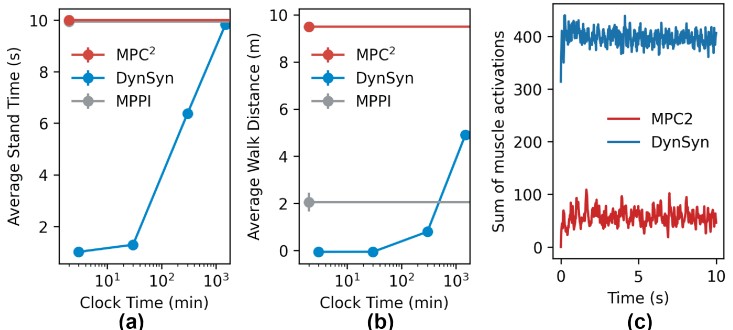

Figure 5: Control performance versus clock time of (a) Stand task, and (b) Walk task. Results show the mean performance with one standard error, averaged over 50 independent trials. (c) Energy consumption during walking.

**Comparison to RL and MPC baselines.** In the Stand and Walk tasks, we compared the control performance of $\mathsf{MPC}^2$ with the current state-of-the-art DRL-based algorithms, DynSyn (He et al., 2024), which identify and utilize muscle synergies to reduce control dimensionality, and demonstrates stable walking control over wholebody musculoskeletal model. We also included the original MPPI (Williams et al., 2016) as a baseline to perform an ablation of our hierarchical pipeline. Figure 5(a)(b) show the total time (training time + deployment time) required for control sequence generation. We observe that DynSyn requires at least one day to achieve effective control in both tasks. While MPPI is capable of maintaining balance in the Stand task, it struggles to generate control sequences for forward movement in the high-dimensional action space. $\mathsf{MPC}^2$ enables stable standing and walking control within 2 minutes, demonstrating a significant time efficiency advantage over DynSyn for deployment. We record the sum of muscle activations as a energy consumption measurement during walking in Figure 5(c).

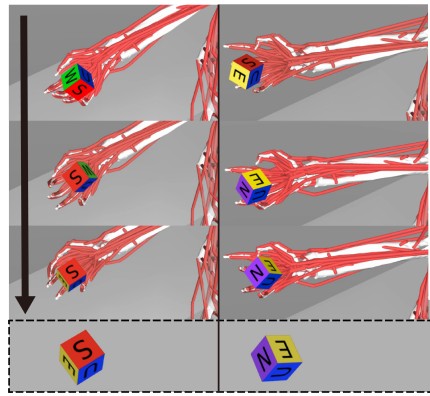

Figure 6: Dexterous manipulation sequences of $\mathsf{MPC}^2$ over arm musculoskeletal model.

Although no energy regularization terms is included in the cost function, we observe that MPC

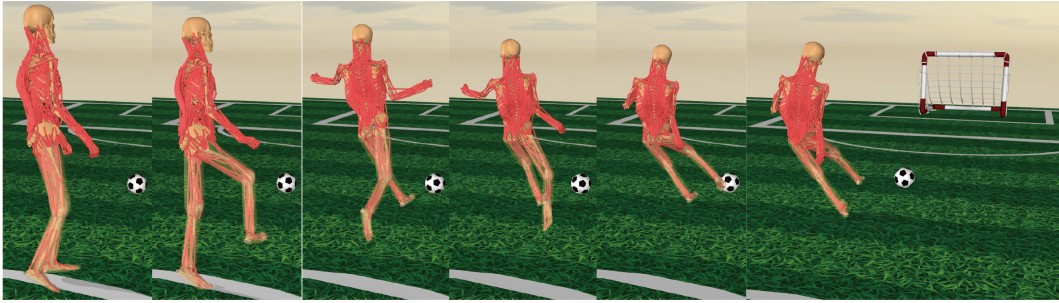

Figure 7: Control sequences of $MPC^2$ in soccer sports imitation. The simulation speed is set to 1% for more frequent planning for rapidly changing motion, where the entire control sequence is learned within 4 minutes.

reduces muscle activation by over 75% compared to DynSyn. We also ran other DRL and MPC baselines in Appendix C.5, where none of them is capable of achieving walking.

## 5.2 CONTROL ACROSS DIFFERENT MORPHOLOGIES

In addition to the challenging task of human full-body locomotion control, we demonstrate that $MPC^2$ generalizes zero-shot to the control of another whole-body musculoskeletal model with a different morphology, as well as to a local human arm model for dexterous manipulation.

**Motion control of ostrich model.** In Figure 3(f), we demonstrate that $MPC^2$ successfully performs morphology computation and achieves stable control for ostrich musculoskeletal models with 120 muscle-tendon units (La Barbera et al., 2021) using the same controller applied to the full-body human model. Notably, the cost function used for the ostrich model is identical to that used for human walking. This highlights $MPC^2$'s ability to perform planning across systems with varying morphologies without requiring training, cost function tuning, or controller parameter adjustments.

**Dexterous manipulation of Arm musculoskeletal model.** In addition to the whole-body movement task, we also evaluate the performance of $MPC^2$ on the dexterous manipulation task. As shown in the Figure 6(a), we need to control a right-hand arm model with 85 muscle-tendon units. By driving the shoulder joint, elbow joint, wrist joint and finger joints, the cube in the hand is adjusted to the specified direction. Compared with whole-body movement, dexterous manipulation requires the controller to have a high reaction speed and more precise control. As shown in Figure 6(b) and Video W28, $MPC^2$ successfully achieves dexterous manipulation of the arm musculoskeletal model and can reach two different target block directions without training, which reflects the generalization of our method to model and control tasks.

## 5.3 AUTOMATIC BEHAVIOR SYNTHESIS WITH OPTIMAL COST FUNCTION DESIGN

The fast control generation speed of $MPC^2$ enables rapid evaluation and iteration of cost function design. In settings where the true objective can be simply described, we can leverage black-box optimization algorithms to discover MPC cost functions that best optimize the true cost function, resulting in *automatic behavior synthesis*. If possible, this functionality is especially crucial in massively multi-task settings, where many complex behaviors must be generated.

We consider this problem in the setting of sports, which often require diverse and complex movements. As a case study, we investigate whether $MPC^2$ combined with a black-box optimizer can automatically learn to kick a soccer ball. We specifically use a Gaussian-process-based Bayesian optimization algorithm to optimize the weights of the position error terms for different body parts (Jones et al., 1998; Rasmussen, 2003; Ament et al., 2023). The optimization objective is the quadratic position error of each body part, with results shown in Figure 10(a). We observe that the cost objective substantially improves compared to the initial settings. Thanks to $MPC^2$'s training-free control generation, our 100 cost design iterations take only around 5 hours, whereas DRL-based methods cannot even complete a single reward evaluation (i.e. a single trained policy) in that time frame. $MPC^2$ successfully imitates the reference trajectory and enables sports motion control, generating

sufficient speed and force to kick the ball (Figure 7). We additionally demonstrate in Appendix C.3 that the automatic cost function design is able to significantly increase the walking speed in both the human and ostrich models.

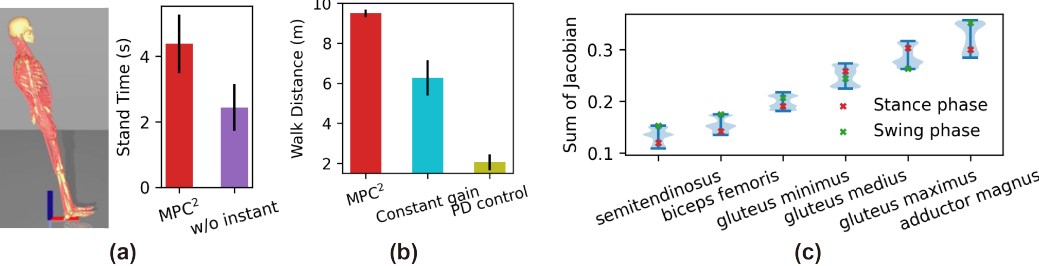

Figure 8: Analysis of $\text{MPC}^2$. Results show the mean performances with one standard error over 20 trials. (a) Control performance of lean backward standing, with initial position shown on the left. Blue axis indicates the vertical direction. (b) Control performance of the Walk task. (c) The distribution of absolute Jacobian summations of a walking trajectory.

## 5.4 ALGORITHM ANALYSIS

To understand superior control performance behind $\text{MPC}^2$ , we investigate both model predictive position controller and morphology-aware proposition controller. The analysis results is shown in Figure 8.

**Instant rollout for rapid planning.** We modified the standing task to evaluate the effectiveness of the instant rollout component in high-level posture planning. As shown in Figure 8(a), instead of starting from an upright position, we set the initial posture of the model to lean significantly backward, requiring a rapid response to recover balance. Our results show that $\text{MPC}^2$ significantly outperforms its variant without the instant rollout, demonstrating that the instant rollout enables a timely response in unstable states.

**Morphology-aware gain design.** We compare $\text{MPC}^2$ with two variants over the low-level actuator controller side: (1) a proportional controller with constant gain settings for all actuators, which has a similar average actuator force as $\text{MPC}^2$, and (2) proportional-derivative (PD) control, setting the derivative gains based on the proportional gains. Figure 8(b) shows that $\text{MPC}^2$ significantly outperforms both the constant gain and PD control variants. In Figure 8(c), we observe that the system's Jacobian effectively identifies the major muscles involved during walking and adapts to different phases of motion. Our morphology-aware gain design automatically prioritizes the major actuators for more efficient control, demonstrating its fidelity in biomechanics.

## 6 CONCLUSION

In this paper, we propose $\text{MPC}^2$, a hierarchical model predictive control method designed to enable near real-time motion control of high-dimensional musculoskeletal systems without the need for training. The algorithm employs a high-level model predictive position controller for posture planning and utilizes a morphology-aware proportional controller to coordinate actuators in achieving the target posture. Using a whole-body model with 700 actuators, we demonstrate the stable control performance of $\text{MPC}^2$ across a wide range of movement tasks, as well as its fast controller generation for efficient cost function optimization. Ablation studies over the algorithm components further verify the principled design and biomechanical fidelity of $\text{MPC}^2$.

## ACKNOWLEDGMENTS

This work is supported by Tsinghua University Initiative Scientific Research Program and STI 2030-Major Projects 2022ZD0209400. Correspondence to: Yanan Sui (ysui@tsinghua.edu.cn).

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

## A  NEURO-MUSCLE DYNAMICS

We use the muscle-tendon units in MuJoCo as our actuator. The input control signal of muscle-tendon units is the neural excitation, denoted as $u$. The muscle activation, denoted as act, is calculated by a first-order nonlinear filter as follows:

$$\frac{\partial \text{act}}{\partial t} = \frac{u - \text{act}}{\tau(u, \text{act})}, \tau(u, \text{act}) = \begin{cases} \tau_{\text{act}} \left(0.5 + 1.5 \cdot \text{act}\right) & u > \text{act} \\ \tau_{\text{deact}} / \left(0.5 + 1.5 \cdot \text{act}\right) & u \leq \text{act} \end{cases}$$

where $\tau_{\text{act}}$ and $\tau_{\text{deact}}$ represent the time constants for activation and deactivation latency, with default values of 10 ms and 40 ms. where $\tau(u, a)$ is the the effective time constant (Millard et al., 2013), which have been smoothed using sigmoid function.

The force produced by a single muscle-tendon unit is given by:

$$f_m(a) = f_{\text{max}} \cdot [F_l(l) \cdot F_v(v) \cdot a + F_p(l)]$$

where $f_{\text{max}}$ is the maximum isometric muscle force, and $a$, $l$, and $v$ represent the activation, normalized length, and normalized velocity of the muscle, respectively. The term $F_p(l)$ accounts for the passive force-length relationship, and the terms $F_l(l)$ and $F_v(v)$ are the force-length and force-velocity functions, which have been fitted using data from biomechanical experiments (Millard et al., 2013).

We use the following 37 major joint positions that determine the whole-body posture: hip (6) knee (2), ankle (2), subtalar (4), spinal (9), shoulder (6), elbow (2), and wrist (6).

The output range of the reinforcement learning policy is typically $[-1, 1]$, and it is then normalized to $[0, 1]$ in order to control the musculoskeletal system. We use the following equation to normalize the action of the policy, which is widely used in MyoSuite environments.

$$a = \frac{1}{1 + e^{-5(a-0.5)}}$$

The reward design is as follows:

$$\text{reward} = \text{reward}_{\text{health}} - \text{cost}_{\text{tasks}}$$

where $reward_{\text{health}}$ is the healthy reward given in each step, We subtract $\text{cost}_{\text{tasks}}$ and add it to $reward_{\text{health}}$ to ensure that the reward remains positive. We find that the original cost weight in the cost function is sufficient for the reinforcement learning algorithm to learn effectively, so we adopt the same weight as used in the cost function.

## B  TASK SETTINGS

The common objective terms are defined as follows:

**Height.** This item limits the difference between the distance from the model's head to its feet and the desired height, encouraging the model to stand.

$$C_{\text{height}} = |H_{\text{head}} - H_{\text{feet}} - H_{\text{target}}|$$

where $H_{\text{head}}$ is the head height, $H_{\text{feet}}$ is the average height of the four feet, and $H_{\text{target}}$ is the target height as a designed parameter for each task

**Upright.** This term encourages the character to maintain an upright posture.

$$C_{\text{upright}} = \left| (1 - \hat{k}_{\text{up}} \cdot \hat{k}_{\text{pelvis}}) + (1 - \hat{k}_{\text{up}} \cdot \hat{k}_{\text{head}}) + 0.1(1 - \hat{k}_{\text{up}} \cdot \hat{k}_{\text{lfoot}}) + 0.1(1 - \hat{k}_{\text{up}} \cdot \hat{k}_{\text{rfoot}}) \right|$$

where $\hat{k}_{\text{up}}$ is the up direction vector of the world coordinate, and $\hat{k}_{\text{head}}$, $\hat{k}_{\text{torso}}$, $\hat{k}_{\text{pelvis}}$, $\hat{k}_{\text{lfoot}}$, $\hat{k}_{\text{rfoot}}$ are the up vectors of the body coordinate for the head, torso, pelvis, left foot, and right foot respectively.

**Balance.** This term encourages keeping the center of mass above the support polygon formed by the feet.

$$C_{\text{balance}} = |\text{COM}_{\text{xy}} - \text{Feet}_{\text{xy}}|$$

where $\text{COM}_{xy}$ is the horizontal projection of the center of mass, and $\text{Feet}_{xy}$ is the average horizontal position of the feet.

**Forward velocity.** This term encourages maintaining a specific forward velocity.

$$C_{\text{vf}} = \left| \mathbf{v}_{\text{COM}} \cdot \hat{k}_{\text{forward}} - v_{\text{target}} \right|$$

where $\mathbf{v}_{\text{COM}}$ is the center of mass velocity, $\hat{k}_{\text{forward}}$ is the forward direction vector, and $v_{\text{target}}$ is the target velocity as a designed parameter for each task.

**Forward angle.** This term discourages sideways motion.

$$C_{\text{vdir}} = \left\| \mathbf{v}_{\text{COM}} - (\mathbf{v}_{\text{COM}} \cdot \hat{k}_{\text{forward}})\hat{k}_{\text{forward}} \right\|_2$$

**Pelvis forward.** This term encourages the character to face forward.

$$C_{\text{bf}} = \left| (1 - \hat{k}_{\text{forward}} \cdot \hat{k}_{\text{pelvis}}) \right|$$

where $\hat{k}_{\text{pelvis}}$ is the forward direction of the pelvis.

**Joint velocity.** This term penalizes excessive joint velocities.

$$C_{\text{jv}} = \|\dot{q}\|_2$$

**Joint position.** This term penalizes extreme joint positions.

$$cost_{jointposition} = \|q\|_2$$

**Feet cross.** This term discourages crossing of the feet and maintains proper leg alignment.

$$C_{\text{fc}} = |\min(0, \hat{k}_{\text{hip}} \cdot \hat{k}_{\text{feet}} - 0.15)$$
$$+ \min(0, \hat{k}_{\text{hip}} \cdot \hat{k}_{\text{toe}} - 0.15) + \min(0, \hat{k}_{\text{hip}} \cdot \hat{k}_{\text{knee}} - 0.15)|$$

where $\hat{k}_{\text{hip}}$, $\hat{k}_{\text{feet}}$, $\hat{k}_{\text{toe}}$, $\hat{k}_{\text{knee}}$ are the direction vector between hip joints, feet centers, toes and knee joints.

The cost function design of each task is as follows:

**Stand**

$$\begin{aligned} C_{\text{stand}} &= 100(C_{\text{height}} + C_{\text{upright}} + C_{\text{balance}}) \\ &\quad + 10C_{\text{vf}} + 10C_{\text{vdir}} + 100C_{\text{bf}} + 0.01C_{\text{jv}} + C_{\text{jp}} \\ H_{\text{target}} &= 1.55 \\ v_{\text{target}} &= 0 \end{aligned}$$

**Walk**

$$\begin{aligned} C_{\text{stand}} &= 100(C_{\text{height}} + C_{\text{upright}} + C_{\text{balance}} \\ &\quad + 10C_{\text{vf}} + 10C_{\text{vdir}} + 100C_{\text{bf}} + 5C_{\text{jp}} + 50C_{\text{fc}} \\ H_{\text{target}} &= 1.55 \\ v_{\text{target}} &= 1 \end{aligned}$$

**Rough**

$$\begin{aligned} C_{\text{stand}} &= 100(C_{\text{height}} + C_{\text{upright}} + C_{\text{balance}} \\ &\quad + 10C_{\text{vf}} + 10C_{\text{vdir}} + 100C_{\text{bf}} + 2C_{\text{jp}} + 50C_{\text{fc}} \\ H_{\text{target}} &= 1.55 \\ v_{\text{target}} &= 0.5 \end{aligned}$$

**Slope**

$$C_{\text{stand}} = 100(C_{\text{height}} + C_{\text{upright}} + C_{\text{balance}}$$
$$+ 10C_{\text{vf}} + 10C_{\text{vdir}} + 100C_{\text{bf}} + 2C_{\text{jp}} + 50C_{\text{fc}}$$
$$H_{\text{target}} = 1.5$$
$$v_{\text{target}} = 0.5$$

**Stair**

$$C_{\text{stand}} = 100(C_{\text{height}} + C_{\text{upright}} + C_{\text{balance}}$$
$$+ 10C_{\text{vf}} + 10C_{\text{vdir}} + 100C_{\text{bf}} + 2C_{\text{jp}} + 50C_{\text{fc}}$$
$$H_{\text{target}} = 1.5$$
$$v_{\text{target}} = 0.5$$
$$C_{\text{stair}} = |\min(H_{\text{forwardfeet}} - H_{\text{stair}}, 0)|$$

where $C_{\text{stair}}$ encourages raising leg before the stair.

For arm musculoskeletal manipulation task, The cost function is adopted from the MPJC Allegro Task[1].

For soccer task, we adopt the same cost function as the MJPC Humanoid Track task[2] without using the control and joint velocity term.

## C    ADDITIONAL EXPERIMENT RESULTS

The video and figures of our experiment is demonstrated in **our project page**.

### C.1    PLANNING UNDER UNCERTAIN MODEL

We conducted an additional experiment to demonstrate that $\text{MPC}^2$ can effectively handle models with uncertainty. Following the implementation in sh MPC, we introduced perturbations to the 'rollout' model during planning by applying Gaussian random forces or torques to each body of the model at every timestep. Figure 9 shows that $\text{MPC}^2$ maintains its control performance until the force standard deviation increases to 8 N or N·m, demonstrating the capability of planning with uncertain model. We consider states from the 'reality' model help correct the errors caused by uncertain model during planning.

### C.2    CONTROL OVER OSTRICH MODELS

In Video W13-W14, we demonstrate that $\text{MPC}^2$ successfully achieves stable control for ostrich musculoskeletal models using the same controller applied to the full-body human model. Notably, the cost function used for the ostrich model is identical to that used for human walking. This highlights $\text{MPC}^2$'s ability to perform planning across systems with varying morphologies without requiring training, cost function tuning, or controller parameter adjustments.

### C.3    AUTOMATIC COST FUNCTION DESIGN

We consider cost function design with $\text{MPC}^2$ is more easier and efficient than reward engineering with DRL for its fast control generation combined with black-box function optimizer. We further demonstrate cost function optimization in both human and ostrich model, with cost function optimization results shown in 10.

As shown in Video W14-W15, starting with same cost function terms and weights as human walking, we utilized Bayesian optimization in weight tuning, improving the walking speed of the human

---

[1] https://github.com/google-deepmind/mujoco_mpc/tree/main/mjpc/tasks/allegro

[2] https://github.com/google-deepmind/mujoco_mpc/blob/main/mjpc/tasks/humanoid/tracking

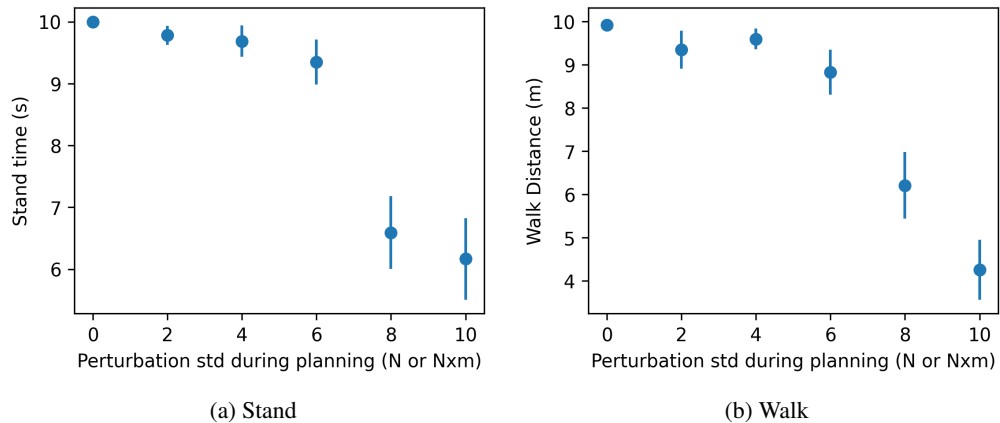

(a) Stand

(b) Walk

Figure 9: Performance of MPC$^2$ under uncertain planning model

from 0.79 m/s to 1.24m/s, and the walking speed of the ostrich from 0.90 m/s to 2.08m/s without manual tuning.

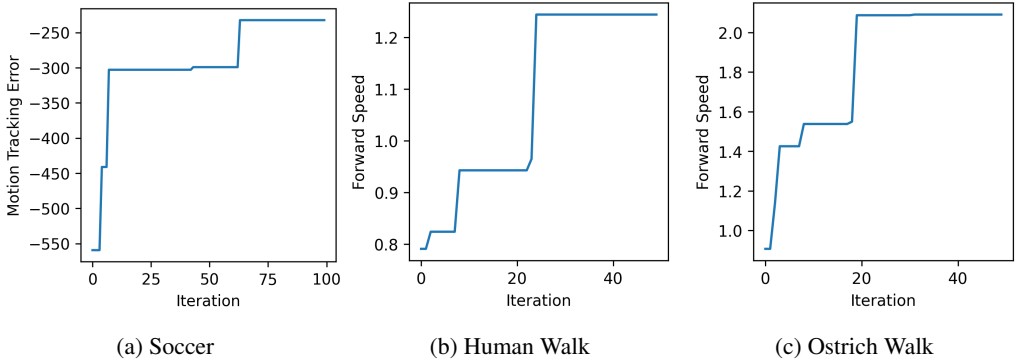

(a) Soccer

(b) Human Walk

(c) Ostrich Walk

Figure 10: Cost function optimization

## C.4   CENTER OF MASS POLYGON SUPPORT

In the Video W16-W17, we plot the centre of mass polygon support during walking for MPC$^2$ and DynSyn. We observe that MPC$^2$ is able to maintain larger polygon support compared to DynSyn, enhancing the stability during walking.

## C.5   PERFORMANCES OF DRL AND MPC BASELINES

We ran MPO, the RL baseline in Wochner et al. (2023), and its succeeded work, DEP-RL (Schumacher et al., 2022), in the standing and walking task. However, as shown in Video W24-W27,we observe that these two method failed to achieve stable standing or running over full-body model with same training steps (5e7) as DynSyn, our DRL baseline in the main paper.

We evaluated six MPC baselines provided by Mujoco MPC, which include both gradient-based methods (Gradient Descent, iLQG, iLQS) and sampling-based methods (Cross Entropy, Robust Sampling, and Sample Gradient). As shown in Video W18-W23 Table 1, none of these methods succeeded in achieving walking with the full-body musculoskeletal model. For non-sampling-based MPC methods, long planning times are required due to the computational demands of deriving the high-dimensional system dynamics, which impedes real-time decision-making. For sampling-based MPC methods, the high-dimensional action space makes it challenging to sample effective control sequences.

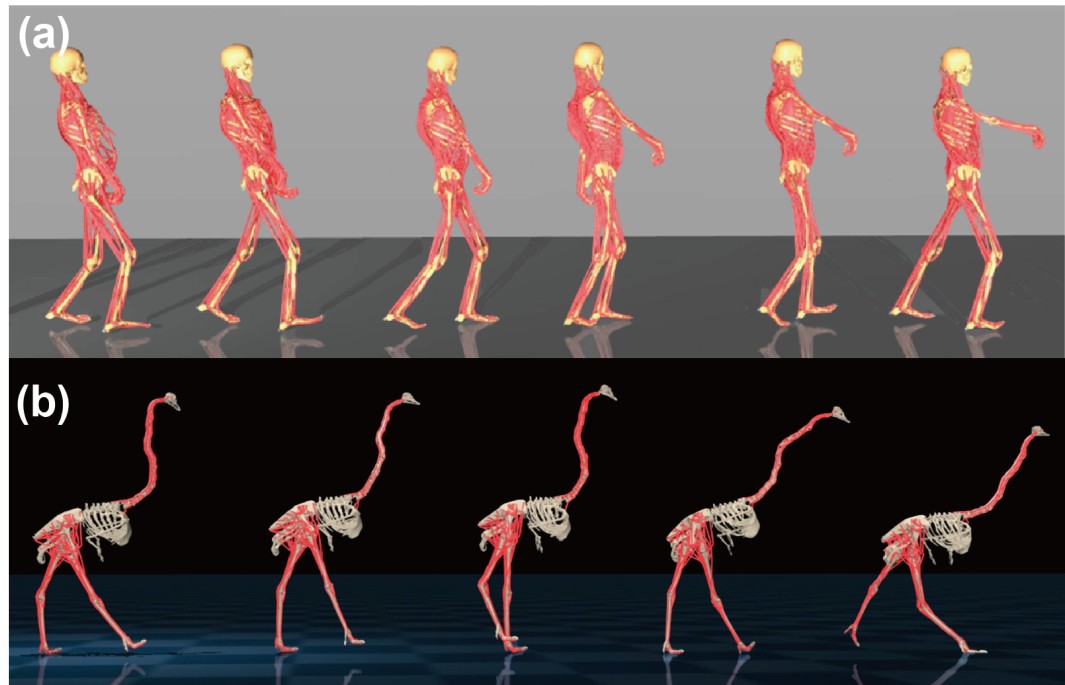

Figure 11: Automatic cost function design for improving the walking speed. (a) Optimized control sequences of $\mathsf{MPC}^2$ over human model. (b) Optimized control sequences of $\mathsf{MPC}^2$ over ostrich model.

| Method | $\mathsf{MPC}^2$ | Gradient Descent | iLQG | MPPI | Cross Entropy | Robust Sampling | Sample Gradient |
|---|---|---|---|---|---|---|---|
| Walk distance | $9.50 \pm 0.19$ | $0.00 \pm 0.00$ | $0.00 \pm 0.00$ | $2.05 \pm 0.40$ | $1.18 \pm 0.23$ | $1.11 \pm 0.15$ | $1.21 \pm 0.24$ |

Table 1: Walking distance of MPC baselines

## D  BASELINES

We compare our algorithm with the reinforcement learning algorithms DynSyn. DynSyn adopt SAC as the basic algorithm and use the DRL framework Stable baselines3. We set control frequency to 10 simulation steps, which can significantly increase the sample efficiency of the reinforcement learning algorithm. All the parameters are reported in the original papers, and we use the same parameters for models with similar complexity. Algorithm hyperparameters are summarized in Table 2.

| Algorithm | Parameter | Task | |
|---|---|---|---|
| | | Stand | Walk |
| SAC | Learning rate | linear schedule(0.001) | |
| | Batch size | 256 | |
| | Buffer size | 1e6 | |
| | Warmup steps | 100 | |
| | Discount factor | 0.98 | |
| | Soft update coeff. | 2 | |
| | Train frequency (steps) | 1 | |
| | Gradient steps | 4 | |
| | Target update interval | 1 | |
| | Environment number | 112 | |
| | Entropy coeff. | auto | |
| | Target entropy | auto | |
| | Policy hiddens | [512, 300] | |
| | Q hiddens | [512, 300] | |
| | Activation | ReLU | |
| | Training steps | 1e7 | |
| DynSyn | Control Amplitude | 5 | |
| | Trajectory steps | 5e5 | |
| | Number of groups | 100 | |
| | aD | 3e7 | |
| | kD | 5e-9 | |

Table 2: Parameters of SAC and DynSyn

