# OpenReview forum: "Motion Control of High-Dimensional Musculoskeletal Systems with Hierarchical Model-Based Planning"
_ICLR.cc/2025/Conference — ICLR 2025 Poster_

### Official Review · Reviewer_Vffr · 2024-10-23

**Soundness:** 3
**Presentation:** 3
**Contribution:** 2
**Rating:** 5
**Confidence:** 3

**Summary:**

This paper presents a system for using a sampling-based model predictive controller to control a musculoskeletal character with a large number of actuators. Key components include: (1) reducing the number of variables using only a single posture target as the control signal. (2) A modification to the sampling strategy that also samples around the current pose.

**Strengths:**

The method is easy to understand and is shown to efficiently generate control signals to do various tasks on a high-dimensional simulated humanoid.

The method allows for the rapid iteration of reward/cost function design.

It is surprising to me that MPPI works well when the sampling signal is just a fixed repeated pose. It could be an interesting observation to the control community.

**Weaknesses:**

There are no videos of the result, so it is hard to judge the quality of the synthesized controller.

There is not much contribution related to machine learning methodology or insights.

**Questions:**

Morphology typically means sizes and shapes of the character, so I was originally expecting to see the controller can adapt to different shapes or sizes, but it seems not the case. I suggest changing the word morphology to something more accurate, e.g., configuration.

I don't think it is fair to compare it to a reinforcement learning method when the action space used is different. It would be better if both algorithms used the same action space.

It seems like instant rollout is only useful for the one very specific scenario discussed. Is it necessary for other scenarios?

It would be great if videos of the results can be shared.

---

> ### Author Response · Authors · 2024-11-20
> **Rebuttal (1/1)**
>
> Thanks for your valuable review. Below we clarify your concerns. Please refer to  **[the anonymous link](https://mpc2-iclr.github.io/MPC2_videos/#actuator_change)** to see our additional experimental results.
>
> >*W1: There are no videos of the result, so it is hard to judge the quality of the synthesized controller.*
> >
> >*Q4: It would be great if videos of the results can be shared.*
>
> Please refer to **[the anonymous link](https://mpc2-iclr.github.io/MPC2_videos/)** to view our video results in the main paper and during the rebuttal.
>
> >*W2: There is not much contribution related to machine learning methodology or insights.*
>
> We believe that the broader implications our hierarchical sampling method are highly relevant to the machine learning, especially our experiments demonstrating that MPC$^2$ enables high-dimensional Bayesian optimization of complex locomotion control. Although deep reinforcement learning methods have traditionally dominated this field, we demonstrate that well-designed control methods combined with online optimization can offer **superior performance while additionally eliminating the need for reward tuning.** This suggests a potential **paradigm shift** in robotics from deep reinforcement learning to predictive control plus black-box optimization, which are all important topics in machine learning. We have added additional experiments on new tasks and morphologies demonstrating that Bayesian optimization can automatically achieve performant behavior with no human intervention.
>
> Our proposed method facilitates planning directly with model morphology without requiring additional training, which is a critical property of embodied intelligence that has yet to be demonstrated by DRL-based methods on complex musculoskeletal systems. Our work achieves motion control of full-body musculoskeletal systems over tasks where no previous method has been successful, such as walking over different terrian conditions and sports imitation. Our methods also demonstrates its robustness to model morphologies, actuator faults, perturbation forces, and uncertain planning models in the additional experiments.
>
> >*Q1: Morphology typically means sizes and shapes of the character, so I was originally expecting to see the controller can adapt to different shapes or sizes, but it seems not the case. I suggest changing the word morphology to something more accurate, e.g., configuration.*
>
> In **[Video W13-W14](https://mpc2-iclr.github.io/MPC2_videos/#ostrich)**, we demonstrate that MPC$^2$ successfully achieves stable control for ostrich musculoskeletal models using the same controller applied to the full-body human model. Notably, the cost function used for the ostrich model is identical to that used for human walking. This highlights MPC$^2$’s ability to perform planning across systems with varying morphologies without requiring training, cost function tuning, or controller parameter adjustments.
>
> >*Q2: I don't think it is fair to compare it to a reinforcement learning method when the action space used is different. It would be better if both algorithms used the same action space.*
>
> As a hierarchical algorithm, MPC$^2$ plans over the target posture and generates actuator controls with the morphology-aware proportional controller. Therefore, the action space of MPC$^2$ is the same as baselines, which is the neural control signal of each muscle actuator.
>
> >*Q3: It seems like instant rollout is only useful for the one very specific scenario discussed. Is it necessary for other scenarios?*
>
> Instant rollout plays a crucial role in enabling the agent to recover to a balanced state when it is falling due to unfavorable samples. Maintaining balance is critical for all the tasks we address, and we use a modified standing task to highlight the significance of this scenario.

---

> > ### Author Response · Authors · 2024-12-01
> >
> > Dear Reviewer Vffr,
> >
> > Thank you for your insightful feedback. Based on your suggestions, we conducted additional experiments and updated both the PDF manuscript and the accompanying anonymous webpage with videos. We hope that our clarifications and the new results across different model morphologies could address your concerns. We look forward to your further feedback.
> >
> > Best regards,
> >
> > Authors of #4268

---

### Official Review · Reviewer_SAwp · 2024-11-04

**Soundness:** 2
**Presentation:** 2
**Contribution:** 2
**Rating:** 6
**Confidence:** 4

**Summary:**

This paper proposes a sampling based hierarchical MPC controller that uses the known model of a simulated high dimensional musculoskeletal humanoid system and achieves good control performance across various tasks without learning a policy representation. This is to be contrasted with deep RL methods that would require a lot of time to train a policy for each task. The hierarchical MPC controller samples target positions that are then used by a morphology aware proportional controller to compute control signals for each DoF. Simulations show that the proposed method works well in various control tasks, where other MPC controllers are claimed to fail invariably.

===== POST-REBUTTAL EDIT =====

I thank the authors for their hard work and the comprehensive rebuttal. The rebuttal has answered some of my concerns and I think the paper could be accepted, it does present a contribution to the simulation/control of high dimensional systems, albeit in a modest way. I raised my score to six as a result.

Here are some more comments based on the rebuttal:
- The github page with extensive results was quite interesting, I am hoping that the authors will integrate the results with the main paper, as I think the main paper lacks the wide coverage of results that the github page presents.
- The gait that is learned with MPC^2 also does not seem very natural. Is that the reason why MPC^2 also fails to stabilize after some time I wonder? How would you learn more natural gaits (from a human walking perspective) with e.g. black-box optimization? This could be mentioned in the discussion/conclusions.
- The paper in the end does not present a learning component. I think the discussion/conclusions should discuss ways to incorporate learning to improve some of the failure modes (gait is not natural, model too complex, MPC not fast enough for real time, more complex scenarios not covered etc.)
- I am still confused about the 'instant rollout' feature, if I understand instant rollout correctly, every MPC algorithm should incorporate instant rollout at some frequency (maybe different from the control frequency) otherwise it wouldn't be corrective. This means that MPC algorithm (running in parallel to the simulation) should always get current states of the robot from the simulation. However from the paper it seems like that is not the default, any reason why that is so? Is it because in simulation you know the model exactly so you can refrain from doing so?

**Strengths:**

The paper shows that a hierarchical and sampling-based implementation of a model-based predictive controller (MPC) can control in simulation a high-dimensional musculoskeletal system for the first time. This is to be contrasted with DRL methods that take a very long time to optimize policies, moreover they need to typically be re-run for each task. In that sense there's a clear contribution to the simulation literature on how to control such high dimensional systems.

**Weaknesses:**

I think the paper has some good contributions to control and simulation of high-dimensional systems as mentioned above, yet there are several weaknesses (some of which can be addressed in the rebuttal phase hopefully):

- There is a very extensive literature on MPC algorithms and analysis, yet the paper does not compare the proposed approach to any other MPC algorithm. Would all of the MPC methods proposed in the literature fail for this complex dynamics case? It is not clear from the text. The tradeoffs of using sampling as opposed to non-sampling based methods are also not discussed.
- I'm not sure if ICLR is the right venue for this paper. Even if we agree that restricting ICLR to papers that 'learn representations' would be too strict, still we're missing a more learning-focus to the paper. This doesn't mean the paper is not well-written or that it is not sound, but I think the paper could benefit more from being targeted to a more controls conference. As of now the learning in the paper is restricted to 'inverting' (without a negative connotation) a known-and-very-complex model. However it would be more interesting for the ML community if the paper would consider not only learning a policy in this particular case, but also extending learning to cover e.g. learning the lower-level policy, learning the higher level policy (if we treat MPPI as a learning method) for different robots, etc.

See also the comments in the Questions section below for more details and questions. Overall I think the paper needs a significant extension of the experiments and ablations in order to justify its conclusions. The heuristic nature of their design decisions (low-level controller, sampling vs non-sampling, effect of sampling size, effect of horizon size, etc.) need to be discussed in more detail and compared against various alternatives. Experiments should also be presented together more concisely, more comprehensively in a table or in a figure (e.g. replacing half-page illustrations of skeletons).

**Questions:**

Some questions and minor comments are given below:

- Missing word in Abstract: e.g. 'it suffers from being computationally-intensive ...'
- Figure 2 is not self explanatory. Also I am confused by the boxes, the 'rollout' box inside big gray box is the access to simulator, which actually replaces the 'reality' of the leftmost box with the skeleton. A more detailed explanation would be appropriate here to ease the introduction.
- From the introduction or the abstract it is not clear how you acquire the 'model' for the Model Predictive Controller that you use. It should be mentioned that you're considering the simulated environment where you have access to / know the model of the robot. In general DRL does not need a model, so the statements should be modified accordingly (RL methods are more general, although sometimes less sample-efficient etc.)
- There does not seem to be any 'learning' in the proposed method, hence it is not clear how to use it in uncertain scenarios, where, e.g., we need to learn a model (or a policy, or both).
- "thus poses a significant bottleneck on the iteration speed of reward engineering." It is not clear up till the sentence what reward engineering is (or what it can be used for) from the introduction, nor the connections of the paper to reward engineering clear.
- Zakka et al. includes DRL in the title, so may not be the right reference for MPC methods vs. DRL in section 2.2.
- Are torque-driven humanoids (Meser et al.) less complex than your system? Section 2.2 does not mention the relation. In fact a 'musculoskeletal' system is not defined.
- notation f is used both for dynamics and for forces (which are part of dynamics, so it's a bit problematic).
- line 160: the cost function used to update MPPI does not seem to be 'cumulative' but only over the horizon of size H.
- line 170: NOT always a local approximation: when H is small, the optimized commands can even cause some systems to be unstable. Moreover you claim f_hat is not equal to f (which is typically the case) hence I think the term "local approximation" is not precise here.
- asynchronized -> asynchronous?
- missing reference for sampling-based MPC methods (there is one reference for an application paper to humanoids)
- again for MPPI, I think the paper is missing a more general reference that introduces the method (the reference is again an application paper it seems)
- \hat{u}^{\theta} notation conflicts with the previous notation where n was used to refer to rollout number.
- I failed to get Algorithm 1, although the text leading up to it was fairly clear. Some terms are not explained, like M_pos, R_MP etc., it's not clear how the initial sigma plays a role, etc.
- The distinction between the terms planning iteration vs. rollout becomes confusing at times, would be nice to define them or illustrate them over the figures.
- the need to introduce instant rollout is not clear to me. why shouldn't the rollout always start from the current state (hence \mu_0 = M_pos(s_0) in your notation). M_pos was explained later in line 276 [and in fact I would simply use p_t as the pose rather than introducing a mask notation, which conflicts with the mass matrix M(q)]. In general this seems to be an re-occurring problem in the paper: properties of certain features of the method are mentioned *before* introducing that feature.
- "While the original Model Predictive Path Integral (MPPI) can be directly employed as a high-level planner for target positions, we find that it lacks the ability to respond quickly to rapidly changing states, such as when the agent is falling. " It wasn't clear to me if this is an argument for the low-level proportional controller? If so why not mention it *after* introducing the low level controller?
- the low level controller seems to be state dependent in eq (11). Hence strictly speaking, it is not a (constant-gain) proportional controller.
- where is the standard error shown in Figure 4?
- shouldn't it be max instead of min in eq. 9?
- it is not clear how eq. (11) resolves the problem of tuning of the gain vector K.
- the design of the low-level controller seems to be very heuristics-driven: one cannot conclude that it will work for arbitrary morphologies/robots, especially given that the results are only given for a particular robot. This seems to be part of a bigger problem, in that, the paper does not propose any learning components to replace DRL or any other state of the art method. This potential lack of generality is worrying, and should be improved.
- "While no previous control methods have demonstrated success in whole-body musculoskeletal systems for these tasks, MPC2 exhibits consistent and stable control performance across various tasks, enabling navigation over different terrain conditions." It would be nice to support this claim by comparing against various MPC/control approaches.
- What are the "DRL-based methods" compared against in section 5.2?
- what are the actual values of the "weights of the position error terms for different body parts" that are optimized? is there an interesting trend, such as significant deviation from uniform values? does it correspond to the last figure in the appendix?
- it's not clear to me what the 'instant rollout' means
- "Figure 6(b) shows that MPC2 significantly outperforms both the constant gain and PD control variants." How are these variants computed? Can you not tune/learn the parameters of the PD (or PID) controller such that it improves the walking distance significantly in Figure 6b?

---

> ### Author Response · Authors · 2024-11-20
> **Rebuttal (1/4)**
>
> Thanks for your detailed review. Below we clarify your concerns. Please refer to  **[the anonymous link](https://mpc2-iclr.github.io/MPC2_videos/#actuator_change)** to see our additional experimental results.
>
> >*W1: The paper does not compare the proposed approach to any other MPC algorithm. The tradeoffs of using sampling as opposed to non-sampling based methods are also not discussed.*
>
> In the main paper, we compared our method with MPPI, a representative sampling-based MPC method that failed to achieve the walking task. Additionally, we evaluated six MPC baselines provided by Mujoco MPC, which include both gradient-based methods (Gradient Descent, iLQG, iLQS) and sampling-based methods (Cross Entropy, Robust Sampling, and Sample Gradient). As shown in **[Figures W18-W23](https://mpc2-iclr.github.io/MPC2_videos/#mpc)**, none of these methods succeeded in achieving walking with the full-body musculoskeletal model.
>
> For non-sampling-based MPC methods, long planning times are required due to the computational demands of deriving the high-dimensional system dynamics, which impedes real-time decision-making. For sampling-based MPC methods, the high-dimensional action space makes it challenging to sample effective control sequences.
>
> Thank you for highlighting the need to discuss non-sampling-based MPC methods. We will include this discussion in the next version of our paper.
>
> >*W2: I'm not sure if ICLR is the right venue for this paper.*
>
> We agree that the hierarchical sampling method alone is not necessarily the best fit for ICLR. However, we believe that the broader implications this method are highly relevant to the conference, especially our experiments demonstrating that MPC$^2$ enables high-dimensional Bayesian optimization of complex locomotion control. Although deep reinforcement learning methods have traditionally dominated this field, we demonstrate that well-designed control methods combined with online optimization can offer **superior performance while additionally eliminating the need for reward tuning.** This suggests a potential **paradigm shift** in robotics from deep reinforcement learning to predictive control plus black-box optimization, which are all topics that are highly relevant to ICLR.
>
> To emphasize the importance of this direction, we have added additional experiments on new tasks and morphologies demonstrating that Bayesian optimization can automatically achieve performant behavior with no human intervention.

---

> ### Author Response · Authors · 2024-11-20
> **Rebuttal (2/4)**
>
> >*Q1: Missing word in Abstract: e.g. 'it suffers from being computationally-intensive ...'*
>
> Thanks for pointing out the typo. We will correct it in the revised version.
>
> >*Q2: Figure 2 is not self explanatory.*
>
> The ‘rollout’ box represents the rollout procedure in MPC$^2$ (as described in Line 4 of Algorithm 1), where sampled control sequences are evaluated, and their corresponding cost function values are computed. Since the model used in the ‘rollout’ box differs from the model depicted in the leftmost ‘reality’ box, we enclose the ‘rollout’ box with dashed frames to highlight this distinction. In our next version, we will revise Figure 2 to enhance its clarity and illustrative quality for improved understanding.
>
> >*Q3: It should be mentioned that you're considering the simulated environment where you have access to / know the model of the robot.*
>
> We have mentioned in the beginning of section 4 (Line 238) that the overall control is conducted in simulation. We will make our statement more clear in the problem setting and our advances compared to DRL-based methods.
>
> >*Q4: it is not clear how to use it in uncertain scenarios*.
>
> We conducted an additional experiment to demonstrate that MPC$^2$ can effectively handle models with uncertainty. Following the implementation in Mujoco MPC, we introduced perturbations to the ‘rollout’ model during planning by applying Gaussian random forces or torques to each body of the model at every timestep. **[Figure W3-W4](https://mpc2-iclr.github.io/MPC2_videos/#uncertain)** show that MPC$^2$ maintains its control performance until the force standard deviation increases to 8 N or N$\cdot$m, demonstrating the capability of planning with uncertain model. We consider states from the 'reality' model help correct the errors caused by uncertain model during planning.
>
> >*Q5: "thus poses a significant bottleneck on the iteration speed of reward engineering." It is not clear up till the sentence what reward engineering is (or what it can be used for) from the introduction, nor the connections of the paper to reward engineering clear*.
>
> Thanks for pointing out the confusing point. We will revise our presentation to make our paper more clear.
>
> >*Q6: Zakka et al. includes DRL in the title, so may not be the right reference for MPC methods vs. DRL in section 2.2.*
>
> Thanks for pointing out the unsuitable reference. we will remove this work in the MPC v.s. DRL sentence.
>
> >*Q7: Are torque-driven humanoids (Meser et al.) less complex than your system? Section 2.2 does not mention the relation. In fact a 'musculoskeletal' system is not defined.*
>
> Yes, we consider the full-body musculoskeletal system we use to be significantly more complex than torque-driven humanoid models. As detailed in HumanoidBench [1], the torque-driven humanoid used by Meser et al. comprises a maximum of 56 actuators with 22 DoF, whereas our musculoskeletal system incorporates 700 actuators with 85 DoF. We introduce the used musculoskeletal system in Section 3.1 and plan to include a direct comparison with torque-driven systems in our next version to further emphasize the complexity of the model.
>
>
> [1] Sferrazza, Carmelo, et al. "Humanoidbench: Simulated humanoid benchmark for whole-body locomotion and manipulation." arXiv preprint arXiv:2403.10506 (2024).
>
> >*Q8: notation f is used both for dynamics and for forces (which are part of dynamics, so it's a bit problematic).*
> >
> >*Q9: line 160: the cost function used to update MPPI does not seem to be 'cumulative' but only over the horizon of size H.*
> >
> >*Q10: line 170: the term "local approximation" is not precise here.*
> >
> >*Q11: asynchronized -> asynchronous?*
> >
> >*Q12: $\hat{u}^{\theta}$ notation conflicts with the previous notation where n was used to refer to rollout number.*
>
> Thanks for pointing out the confusing notation and unprecise statement, and we will revise in the next version.
>
> >*Q13 & Q14: missing reference for sampling-based MPC and MPPI methods*
>
>
> Thanks for pointing out the missing references, and we will conduct more comprehensive literature search and include the relevant works in the next version.
>
> >*Q15: I failed to get Algorithm 1, although the text leading up to it was fairly clear. Some terms are not explained, like $M_{pos}, R_{MP}$ etc., it's not clear how the initial sigma plays a role, etc.*
> >
> >*Q16: The distinction between the terms planning iteration vs. rollout becomes confusing at times, would be nice to define them or illustrate them over the figures.*
>
> $M_{pos}$ was introduced in line 276, and $R_{MP}$ was introduced in e.q. (8). We will improve our presentation in the text and figures to reduce confusions in the next version.

---

> ### Author Response · Authors · 2024-11-20
> **Rebuttal (3/4)**
>
> >*Q17: the need to introduce instant rollout is not clear to me.*
>
> The “target posture” planned by MPC$^2$ and the agent’s current posture are typically not identical, although they share the same dimensionality. In our approach, MPPI begins with the nominal policy optimized in the previous iteration, which corresponds to the “target posture” from the last round ($\mu$ in the inputs of Algorithm 1).
>
> The instant rollout is sampled directly from the model's current posture. we consider additional planning from the current posture helps to better find policies to rapid response to the changing state, as evidenced by our ablation study in section 5.3.
>
> Thanks to pointing out the confusing point, and we will improve our presentation in the next version.
>
> >*Q18: "While the original Model Predictive Path Integral (MPPI) can be directly employed as a high-level planner for target positions, we find that it lacks the ability to respond quickly to rapidly changing states, such as when the agent is falling. " It wasn't clear to me if this is an argument for the low-level proportional controller? If so why not mention it after introducing the low level controller?*
>
> This is an argument for high-level controller.  In scenarios such as where the agent is falling, the current state of the agent can deviate significantly from the planned target posture. In such cases, it becomes challenging for MPPI to sample a feasible policy capable of recovering the agent to a balanced state.
>
> >*Q19: the low level controller seems to be state dependent in eq (11). Hence strictly speaking, it is not a (constant-gain) proportional controller.*
>
> Yes, the low-level controller is state-dependent, with its gains adaptively adjusted based on the model’s posture (represented by the Jacobian) and the deviation between the current and target postures.
>
> >*Q20: where is the standard error shown in Figure 4?*
>
> For both MPC$^2$ and DynSyn, the standard error is so small (<0.06) that it is not visually discernible, being fully obscured by the dots in the graph. We will improve the presentation to better demonstrate our results.
>
> >*Q21: shouldn't it be max instead of min in eq. 9?*
>
> Muscle force is defined as a contractile force, which is represented as negative in Mujoco’s muscle dynamics. Consequently, we use the min operator to ensure that only negative values are considered in our computations.
>
> >*Q22: it is not clear how eq. (11) resolves the problem of tuning of the gain vector K.*
>
> The first term is the sum of the absolute Jacobian values, which represents the influence of actuators on joints. As shown in Fig. 6(c), the use of absolute Jacobian summations effectively identifies the primary actuators associated with a given model posture.
>
> The second term represents the difference between the current and target postures, which mitigates oscillations caused by proportional control.
>
> Our approach allows the proportional gains to be automatically adjusted based on both the system morphology (via the Jacobian) and the distance to the target posture (the difference between current and target posture).
>
> >*Q23: the design of the low-level controller seems to be very heuristics-driven: one cannot conclude that it will work for arbitrary morphologies/robots, especially given that the results are only given for a particular robot. This seems to be part of a bigger problem, in that, the paper does not propose any learning components to replace DRL or any other state of the art method. This potential lack of generality is worrying, and should be improved.*
>
> In **[Video W13-W14](https://mpc2-iclr.github.io/MPC2_videos/#ostrich)**, we demonstrate that MPC$^2$ successfully achieves stable control for ostrich musculoskeletal model using the same controller applied to the full-body human model. Notably, the cost function used for the ostrich model is identical to that used for human walking. This highlights MPC$^2$’s ability to perform planning across systems with varying morphologies without requiring training, cost function tuning, or controller parameter adjustments.
>
> >*Q24: "While no previous control methods have demonstrated success in whole-body musculoskeletal systems for these tasks, MPC2 exhibits consistent and stable control performance across various tasks, enabling navigation over different terrain conditions." It would be nice to support this claim by comparing against various MPC/control approaches.*
>
> In response to the W1, we provide evidence that a wide range of MPC baselines failed to achieve walking control with the full-body musculoskeletal model.

---

> ### Author Response · Authors · 2024-11-20
> **Rebuttal (4/4)**
>
> >*Q25: What are the "DRL-based methods" compared against in section 5.2?*
>
> We compared to the training time of DynSyn[2], which requires about 16 hours to train 5e7 steps. MPO[3] and DEP-RL[4] in our additional experiment require around 20 hours to train the same steps.
>
>
> [2] He, Kaibo, et al. "DynSyn: dynamical synergistic representation for efficient learning and control in overactuated embodied systems." arXiv preprint arXiv:2407.11472 (2024).
>
> [3] Wochner, Isabell, et al. "Learning with muscles: Benefits for data-efficiency and robustness in anthropomorphic tasks." Conference on Robot Learning. PMLR, 2023.
>
> [4] Schumacher, Pierre, et al. "Dep-rl: Embodied exploration for reinforcement learning in overactuated and musculoskeletal systems." arXiv preprint arXiv:2206.00484 (2022).
>
> >*Q26: what are the actual values of the "weights of the position error terms for different body parts" that are optimized? is there an interesting trend, such as significant deviation from uniform values? does it correspond to the last figure in the appendix?*
>
> Yes, the last figure in the appendix illustrates the optimized weights (pink) compared to the default settings (gray) in the Mujoco MPC humanoid tracking task. The original cost function comprises 19 terms, including both position and velocity tracking components. For optimization, we extracted the position error terms for nine body parts while keeping the remaining terms fixed at their default weights.
>
> >*Q27: it's not clear to me what the 'instant rollout' means*
>
> The ‘instant rollout’ refers to sampling new target postures directly from the model’s current posture, rather than from the target posture optimized in the previous iteration.
>
> >Q*28: "Figure 6(b) shows that MPC2 significantly outperforms both the constant gain and PD control variants." How are these variants computed? Can you not tune/learn the parameters of the PD (or PID) controller such that it improves the walking distance significantly in Figure 6b?*
>
> For the variant of the proportional controller with a constant gain design, we assign a constant gain to every actuator to ensure the controller produces a similar average actuator force as the morphology-aware gain design.
>
> For the variant of the PD controller, we incorporate an actuator velocity difference term and follow the implementation in [5] to set the derivative gains based on the morphology-aware proportional gain ($k_d = 2\sqrt{k_p}$). We also experimented with other gain designs for the PD controller, but these attempts resulted in degraded performance, likely due to the characteristics of the first-order muscle actuators in Mujoco.
>
> We consider adaptively adjusting actuator gains based on the model’s morphology contributes to stable and efficient control by prioritizing the use of actuators with the greatest influence in achieving the target posture. As demonstrated in Figure 6(c) in the main paper, using the sum of absolute Jacobian values effectively identifies the major muscles engaged during different phases of walking.
>
> [5] Feng, Yusen, Xiyan Xu, and Libin Liu. "MuscleVAE: Model-Based Controllers of Muscle-Actuated Characters." SIGGRAPH Asia 2023 Conference Papers. 2023.

---

> ### Author Response · Authors · 2024-11-26
>
> Thank you for raising your score! We appreciate your constructive feedback and will continue to improve our work based on your suggestions. Below, we address your additional comments.
>
> > C1: The github page with extensive results was quite interesting, I am hoping that the authors will integrate the results with the main paper, as I think the main paper lacks the wide coverage of results that the github page presents.
>
> Thank you for your suggestions. We will incorporate the additional results into the main paper in our next version.
>
> > C2: The gait that is learned with MPC$^2$ also does not seem very natural. Is that the reason why MPC$^2$ also fails to stabilize after some time I wonder? How would you learn more natural gaits (from a human walking perspective) with e.g. black-box optimization? This could be mentioned in the discussion/conclusions.
>
> We think the main reason for the unnatural gait is the absence of reference trajectories or terms that regularize the behavior of MPC$^2$. Currently, MPC$^2$ focuses solely on maintaining balance and moving forward based on the given cost function. To achieve a more natural gait, we can incorporate additional terms to account for fatigue or energy consumption, which promotes energy-efficient movements that often result in more natural behavior. Combining black-box optimization with MPC$^2$, we can automatically and efficiently optimize the cost weights, enhancing both the naturalness of the gait and the overall control performance.
>
> >C3: The paper in the end does not present a learning component. I think the discussion/conclusions should discuss ways to incorporate learning to improve some of the failure modes (gait is not natural, model too complex, MPC not fast enough for real time, more complex scenarios not covered etc.)
>
> Thanks for your suggestions. We've emphasized the Bayesian optimization part in the current PDF. We will add additional discussions of how to incorporate learning components into MPC$^2$ in the next version.
>
> >C4: I am still confused about the 'instant rollout' feature, if I understand instant rollout correctly, every MPC algorithm should incorporate instant rollout at some frequency (maybe different from the control frequency) otherwise it wouldn't be corrective. This means that MPC algorithm (running in parallel to the simulation) should always get current states of the robot from the simulation. However from the paper it seems like that is not the default, any reason why that is so? Is it because in simulation you know the model exactly so you can refrain from doing so?
>
> During the algorithm implementation, we reviewed the MPC implementations in Mujoco MPC and other repositories and observed that they did not include this correctness term. This is likely because these implementations plan directly over the actuator space rather than a high-level target space. Directly planning over the actuator space allows the exact same control signals to be applied to the model, eliminating the need for correctness adjustments based on the current model state.
> However, when planning over high-level targets, deviations between the target and the current model state are common, necessitating instant rollouts to ensure correctness and alignment with the actual state of the system.

---

### Official Review · Reviewer_VP3V · 2024-11-04

**Soundness:** 2
**Presentation:** 3
**Contribution:** 3
**Rating:** 8
**Confidence:** 4

**Summary:**

## Summary:
The paper introduces a control algorithm called Model Predictive Control with Morphology-aware Proportional Control (MPC2) to address
 challenges associated with the real-time control of high-dimensional musculoskeletal systems. The problem is complex due to the inherent complexity of over-actuated nature involving large state and action spaces. The authors proposed a Sampling based MPC with Morphology-aware Proportional Controller for actuator co-ordination (MPC2). They show that the methodology can produce stable motions without any training/data-driven steps, resulting in zero-shot motion control. They used Deep learning based approach as a baseline to compare their method for various tasks such as standing, walking on flat/rough surface, slope, stairs and kicking a ball.

## Methodology:
MPC2 combines two main strategies:
1. High-Level Planning via Model Predictive Control (MPC): This component of MPC2 uses a sampling-based model predictive controller to plan target postures for the system. It operates by forecasting the desired postures over a finite horizon and computing optimal control actions to achieve these postures.
2. Low-Level Control via Morphology-aware Proportional Control: This controller works at the actuator level, coordinating the actuators to reach the target joint positions determined by the high-level planner. It adjusts the actuators dynamically based on the morphology of the system, ensuring efficient and effective movement control.

## Key Contributions and Findings:
1. Training-Free Control: Unlike Deep Reinforcement Learning based methods that require extensive training, MPC2 enables zero-shot control, meaning it can control the system effectively without any prior training on specific tasks.
2. Adaptability to Diverse Tasks: The algorithm was tested on various movement tasks, including standing, walking on different terrains, and sports motion imitation. These experiments demonstrated its capability to adapt to a wide range of activities without the need for manual tuning or reward engineering.
3. Real-Time Performance: One of the significant advancements of MPC2 is its ability to perform near real-time control. This is due to their Hierarchical control strategy, reducing of the action space for MPC (dz = 37) instead of the full control over all 700 tendons.
4. Efficient Optimization: The paper shows that MPC2 can optimize its cost function design automatically through Bayesian optimization. This feature significantly reduces the manual effort typically needed in traditional control systems.

## Experimental Validation:
The effectiveness of MPC2 was validated through a series of experiments conducted on a simulated 700-actuator musculoskeletal model. The model was tested across various scenarios that require high levels of coordination and control precision. Results from these experiments confirm the robustness and versatility of MPC2, outperforming existing deep reinforcement learning-based methods in terms of speed and adaptability. Their website provides video demonstrations of MPC2’s real-time control capabilities across various movement tasks. These videos effectively illustrate the adaptability and robustness of the proposed algorithm. They show MPC2 handling complex movements like walking on uneven terrain, climbing stairs, and executing sports motions (e.g., kicking a ball) with stable, smooth coordination across multiple actuators. These visual results add strong qualitative evidence to the paper’s claims about MPC2’s ability to perform zero-shot control without training and adapt to different motion tasks with high precision and reliability.

## Conclusion:
The development of MPC2 marks a significant step forward in the field of robotics and biomechanics, offering a powerful tool for the high-dimensional control of complex systems. This method holds potential not only for improving robotic system performance but also for advancing human-related applications such as prosthetics and rehabilitation robotics, where adaptive, real-time control is crucial.

**Strengths:**

The paper definitely presents some key strengths:
1. Originality: While breaking down the high-dimensional control/planning problem into multiple stages is a very know concept, the paper presented first of its kind to integrate MPPI with Morphology Aware controller. The paper thus provides a new solution for high dimensional motion control.
2. Quality: The paper provides a thorough experimental validation in Mujoco simulation. On technical aspect, the use of Bayesian optimization for automated cost function tuning streamlined the traditionally labor-intensive process of reward engineering.
3. Clarity: The paper is a very well written providing curriculum approach to MPC. The provided code is well structured and documented.
4. Significance: Though the results were presented only in simulation and qualitatively seemed far from applicable to real applications, the approach presented has significant practical impact on real-time controls.

**Weaknesses:**

I found several weak points in relation to the experiments and validation of the proposed method (it could be also a result of my background from classical control theory).

1. Lack of quantitative stability metrics: The paper lacks in presenting the performance of their approach on key stability metrics used across the humanoids community like centre of mass polygon support, energy efficiency, etc. Additionally, integrating these metrics into the MPC cost function could enhance the results significantly. (De Viragh, Yvain, et al. "Trajectory optimization for wheeled-legged quadrupedal robots using linearized zmp constraints." IEEE Robotics and Automation Letters 4.2 (2019): 1633-1640.)

2. Experimental results:
    1. Lacks quantitative discussion in comparison with RL based methods (the stability/energy metrics would have helped here).
    2. The paper’s comparative analysis primarily uses a DRL-based baseline, but the implementation quality of this baseline appears to be lacking. The RL approach (Learning with Muscles: Benefits for Data-Efficiency and Robustness in Anthropomorphic Tasks) should have been compared.

3. Figure 5.a. mentions "Best Objective Value" but it isnt discussed in the paper, what that means and why it is significant.

**Questions:**

1. Could the authors provide additional insights into the computational requirements for real-time applications of MPC2?
2. Could you make the experimental more elaborative towards quantitative results ?

**Details Of Ethics Concerns:**

No Concerns.

---

> ### Author Response · Authors · 2024-11-20
> **Rebuttal (1/1)**
>
> Thanks for your appreciations of our work! Below we clarify your concerns. Please refer to  **[the anonymous link](https://mpc2-iclr.github.io/MPC2_videos/#actuator_change)** to see our additional experimental results.
>
> >*W1: Lack of quantitative stability metrics.*
> >
> >*W2.1: Lacks quantitative discussion in comparison with RL based methods (the stability/energy metrics would have helped here).*
> >
> >*Q2: Could you make the experimental more elaborative towards quantitative results?*
>
> Thank you for your suggestions of incorporating stability metrics into cost function design. We did minimize the distance between the CoM xy position and the center of feet xy position to maintain balance for the model. Below we demonstrate some quantitative results for MPC and DynSyn (the only DRL baseline that can achieve walking).
>
>
> In the **[Video W16-W17](https://mpc2-iclr.github.io/MPC2_videos/#com)**, we plot the centre of mass polygon support during walking for MPC$^2$ and DynSyn. We observe that MPC$^2$ is able to maintain larger polygon support compared to DynSyn, enhancing the stability during walking.
>
> We also record the sum of muscle activations as a energy consumption measurement during walking in **[Figure W7](https://mpc2-iclr.github.io/MPC2_videos/#energy)**. Although no energy regularization terms is included in the cost function, we observe that MPC reduces muscle activation by over 75% compared to DynSyn.
>
> >*W2.2: The paper’s comparative analysis primarily uses a DRL-based baseline, but the implementation quality of this baseline appears to be lacking.*
>
> Thank you for pointing out the potential baselines. We ran MPO in the mentioned paper, and its succeeded work, DEP-RL [1], in the standing and walking task. However, as shown in **[Video W24-W27](https://mpc2-iclr.github.io/MPC2_videos/#drl)**,we observe that these two method failed to achieve stable standing or running over full-body model with same training steps (5e7) as DynSyn, our DRL baseline in the main paper.
>
> [1] Schumacher, Pierre, et al. "Dep-rl: Embodied exploration for reinforcement learning in overactuated and musculoskeletal systems." arXiv preprint arXiv:2206.00484 (2022).
>
> >*W3: Figure 5.a. mentions "Best Objective Value" but it isnt discussed in the paper, what that means and why it is significant.*
>
> The term "Best Objective Value" denotes the best objective function value across trials during cost function optimization. In the sports imitation task, we set the sum of quadratic position error of each body part over whole task horizon as the objective function. In this way we can penalize large posture deviation from the reference trajectory.
>
> >*Q1: Could the authors provide additional insights into the computational requirements for real-time applications of MPC2?*
>
> The primary computational bottleneck of MPC$^2$ lies in the calculation of forward dynamics, which is necessary both for evaluating rollout performance (a requirement shared with other sampling-based MPC methods) and for determining target actuator lengths. We believe the efficiency of MPC$^2$ can be further improved through:
> 1.	Advanced hardware to accelerate the computation of exact forward dynamics, or
> 2.	A well-calibrated approximate model. In **[Figure W3-W4](https://mpc2-iclr.github.io/MPC2_videos/#uncertain)** we demonstrate that MPC$^2$ is capable of handling uncertain models, highlighting its robustness in scenarios involving model approximations.

---

### Official Review · Reviewer_4m7j · 2024-11-04

**Soundness:** 3
**Presentation:** 3
**Contribution:** 2
**Rating:** 6
**Confidence:** 3

**Summary:**

The paper presents a hierarchical model predictive control approach for a  700-dimensional musculoskeletal system, where a modified version of MPPI is used as a higher-level controller to output target postures that are tracked by a lower-level proportional controller. The gains of the lower-level proportional controller are set using a morphology-aware function based on the Jacobian matrices of the current state and the target posture. The method is evaluated on several tasks and the paper reports ablations of the method.

**Strengths:**

- The presented method is simple, can handle the high dimensionality of the action space associated with muscle-actuated systems, and performs well on several challenging tasks, such as walking on rough terrain, walking on slopes, or over stairs.
- Fast cost-design iteration time to evaluate/generate new controllers compared to Deep RL methods.
- The paper is well-written overall and the ablation studies justify different parts of the proposed method.

**Weaknesses:**

- I have concerns that the conference might not be a good fit for the paper. I struggle to classify the presented hierarchical sampling-based approach for control as an area or subfield of machine learning. Especially, considering that MPPI and many of its variants have been traditionally presented in robotics conferences. If the authors can clarify, how the conference is a good fit for the paper or if this is not a concern for any other reviewer, I will not oppose the acceptance of this paper in an exercise of  " broadening definitions of originality and significance".

- The paper would benefit from making a clear distinction between the iteration time to create a controller and the time needed to execute a controller at test time. L121 states “Compared to DRL, model predictive control allows for real-time control”.  It is well known that training DRL policies can be time-consuming, however, after the training stage, most DRL policies can be queried/executed in the order of milliseconds thus effectively allowing for real-time control, unlike the presented method which is only suitable for near real-time deployment/execution.


- Probably due to the challenging nature of the setting, only a few baselines are provided and the performance of the baselines is not very competitive.  AHAC [1] might provide a stronger baseline for the comparisons.  AHAC has also been tested on a muscle-actuated humanoid task and has better performance than SAC and PPO on such tasks.
   - [1] Ignat Georgiev et al.  Adaptive Horizon Actor-Critic for Policy Learning in Contact-Rich Differentiable Simulation.

**Questions:**

- Apart from only being near real-time at execution time? what are other limitations of the method?
- Is the method robust to perturbations? Can the humanoid keep walking when unknown external forces are applied?
- Is a different cost function used to obtain a controller for each task? If so, is there a systematic way to unify/merge cost functions to obtain a controller that can perform multiple tasks at the same time?
- One of the motivations for musculoskeletal systems is its robustness in case of actuator failure. How does the method’s performance degrade in the presence of actuator failures  (when only a subset of tendons can be actuated)? Such evaluation would further highlight the benefits of the presented approach.

---

> ### Author Response · Authors · 2024-11-20
> **Rebuttal (1/2)**
>
> Thanks for your insightful review. Below we clarify your concerns. Please refer to **[the anonymous link](https://mpc2-iclr.github.io/MPC2_videos/#actuator_change)** to see our additional experimental results.
>
> >*W1: I have concerns that the conference might not be a good fit for the paper.*
>
> We agree that the hierarchical sampling method alone is not necessarily the best fit for ICLR. However, we believe that the broader implications this method are highly relevant to the conference, especially our experiments demonstrating that MPC$^2$ enables high-dimensional Bayesian optimization of complex locomotion control. Although deep reinforcement learning methods have traditionally dominated this field, we demonstrate that well-designed control methods combined with online optimization can offer **superior performance while additionally eliminating the need for reward tuning.** This suggests a potential **paradigm shift** in robotics from deep reinforcement learning to predictive control plus black-box optimization, which are all topics that are highly relevant to ICLR.
>
> To emphasize the importance of this direction, we have added additional experiments on new tasks and morphologies demonstrating that Bayesian optimization can automatically achieve performant behavior with no human intervention.
>
> >*W2: The paper would benefit from making a clear distinction between the iteration time to create a controller and the time needed to execute a controller at test time.*
>
> Thanks for your advice, and we will clarify the training and deployment time in our next version.
>
> >*W3: Probably due to the challenging nature of the setting, only a few baselines are provided and the performance of the baselines is not very competitive. AHAC [1] might provide a stronger baseline for the comparisons.*
> >
> >*[1] Ignat Georgiev et al. Adaptive Horizon Actor-Critic for Policy Learning in Contact-Rich Differentiable Simulation.*
>
> Thanks for pointing out the potential baseline. We read the paper of AHAC and found it is not applicable at the moment to deploy the full-body musculoskeletal model on the used differentiable physics engine dFlex, as it does not provide a direct interface to compute non-linear muscle dynamics like in Mujoco. Besides, MPC$^2$ does not require the assumption that the dynamics is differentiable.
>
> Following the suggestion of reviewer VP3V, we conducted additional experiments with two DRL-based baselines, MPO [1] and DEP-RL [2], both of which have previously demonstrated success in controlling musculoskeletal systems in Mujoco. However, as shown in **[Video W24-W27](https://mpc2-iclr.github.io/MPC2_videos/#drl)**,we observe that these two method failed to achieve stable standing or running over full-body model with same training steps (5e7) as DynSyn, our DRL baseline in the main paper.
>
> [1] Wochner, Isabell, et al. "Learning with muscles: Benefits for data-efficiency and robustness in anthropomorphic tasks." Conference on Robot Learning. PMLR, 2023.
>
> [2] Schumacher, Pierre, et al. "Dep-rl: Embodied exploration for reinforcement learning in overactuated and musculoskeletal systems." arXiv preprint arXiv:2206.00484 (2022).

---

> ### Author Response · Authors · 2024-11-20
> **Rebuttal (2/2)**
>
> >*Q1: Apart from only being near real-time at execution time? what are other limitations of the method?*
>
> We consider that planning dimensionality remains a limitation of our approach. Our findings suggest that planning based on one or two postures is sufficient to generate effective control sequences for walking and standing. However, extending planning to a sequence of postures introduces high-dimensionality challenges that hinder sampling effective control. This limitation restricts MPC$^2$ from achieving complex movements that require rapid changes over short time intervals.
>
> >*Q2: Is the method robust to perturbations? Can the humanoid keep walking when unknown external forces are applied?*
>
> Yes, MPC$^2$ demonstrates robustness to certain perturbations, as evidenced in two walking scenarios. As shown in **[Video W11 and W12](https://mpc2-iclr.github.io/MPC2_videos/#force)**, MPC$^2$ successfully maintains forward walking despite the application of significant external forces, including large, random, short-term forces (500N applied for 0.2 seconds every 1 second) and consistent, random forces (100N applied continuously). These results highlight the system’s ability to adapt and maintain stability under challenging conditions.
>
> >*Q3: Is a different cost function used to obtain a controller for each task? If so, is there a systematic way to unify/merge cost functions to obtain a controller that can perform multiple tasks at the same time?*
>
> We employ the same cost function for walking across various scenarios, including flat floors, rough terrain, and slopes. Designed to ensure stability and forward motion, this cost function is also directly applicable to ostrich walking without any modifications, as shown in **[Video W13](https://mpc2-iclr.github.io/MPC2_videos/#ostrich)**. We have demonstrated that our cost function can successfully handle multiple tasks across different models simultaneously.
>
> We consider cost function design with MPC$^2$ is more easier and efficient than reward engineering with DRL for its fast control generation combined with black-box function optimizer. We further demonstrate cost function optimization in both human and ostrich model.
>
> In **[Figure W5-W6 and Video W14-W15](https://mpc2-iclr.github.io/MPC2_videos/#cost_design)**, starting with same cost function terms and weights as human walking, we utilized Bayesian optimization in weight tuning, improving the walking speed of the ostrich from 0.90 m/s to 2.08m/s, and the walking speed of the human from 0.79 m/s to 1.24m/s without manual tuning.
>
> >*Q4: One of the motivations for musculoskeletal systems is its robustness in case of actuator failure. How does the method’s performance degrade in the presence of actuator failures (when only a subset of tendons can be actuated)? Such evaluation would further highlight the benefits of the presented approach.*
>
> Yes, we demonstrate that MPC$^2$ effectively leverages the over-actuated nature of musculoskeletal systems to achieve stable control even in the presence of actuator failures. As illustrated in **[Video W10](https://mpc2-iclr.github.io/MPC2_videos/#actuator_change)**, MPC$^2$ dynamically adapts its control strategy to maintain forward walking despite the sudden disablement of the posterior muscles in the right leg. We also find trained DRL agent fails to walk with actuator faults, as shown in the **[Video W3](https://mpc2-iclr.github.io/MPC2_videos/#dynsyn_fail)**.

---

> > ### Comment · Reviewer_4m7j · 2024-11-28
> > **Response.**
> >
> > Thanks for the clarifications. Considering that the additional experiments have strengthened the paper, I have adjusted my score accordingly.

---

> > > ### Author Response · Authors · 2024-11-28
> > >
> > > Thank you for your feedback! We will continue to improve the paper based on your suggestions.

---

### Author Response · Authors · 2024-11-25

Dear Reviewers,

Thank you again for your valuable feedback on our submission. We greatly appreciate the time and effort you took to review our work on optimizing the control of high-dimensional embodied systems.

We have provided additional results and detailed explanations in the rebuttal comments to address the points you raised. With the deadline approaching in less than two days, we hope that you will provide additional feedback to ensure that we have fully addressed all of your concerns.

Thank you for your continued support!

Best regards,

Authors of #4268

---

### Meta-Review · Area_Chair_yUnA · 2024-12-24

**Metareview:**

The paper presents Model Predictive Control with Morphology-aware Proportional Control (MPC), a hierarchical model-based algorithm enabling zero-shot, near real-time control of high-dimensional nonlinear systems, demonstrated on a 700-actuator musculoskeletal model across diverse movement tasks without requiring training or extensive reward engineering.

The reviewers are generally supportive of accepting this paper, acknowledging its contributions, including (1) its simplicity paired with the ability to handle high-dimensional action spaces, (2) adaptability to diverse tasks, (3) efficiency, and (4) clear presentation.

During the Author-Reviewer Discussion phase, the authors provided thorough responses that successfully convinced some reviewers to raise their scores. Most reviewers are supportive of accepting the paper, with the exception of Reviewer Vffr, who did not engage in either the Author-Reviewer Discussion or Reviewer Discussion phases. Consequently, the AC has decided to reduce the weight of this review.

Still, the AC recommends that the authors carefully revisit both the original and post-rebuttal reviewer comments to address remaining concerns in a revised version of the paper.

**Additional Comments On Reviewer Discussion:**

In the Reviewer Discussion phase, despite Reviewer Vffr expressing a negative opinion of the paper, they did not engage in the discussion. As a result, the majority opinion remained unchanged.

---

### Decision · Program_Chairs · 2025-01-22

Accept (Poster)